# Multifaceted analysis of cross-tissue transcriptomes reveals phenotype–endotype associations in atopic dermatitis

Aiko Sekita [1,2], Hiroshi Kawasaki[1,2], Ayano Fukushima-Nomura [2], Kiyoshi Yashiro[2], Keiji Tanese[2], Susumu Toshima[1,2], Koichi Ashizaki[1,2,3], Tomohiro Miyai [1,2], Junshi Yazaki [1], Atsuo Kobayashi[1], Shinichi Namba [4,5], Tatsuhiko Naito [4], Qingbo S. Wang[1,4,5], Eiryo Kawakami [3,6], Jun Seita [1,3], Osamu Ohara [7], Kazuhiro Sakurada[3,8], Yukinori Okada [1,4,5] ✉, Masayuki Amagai [1,2] ✉ & Haruhiko Koseki [1,9] ✉

Atopic dermatitis (AD) is a skin disease that is heterogeneous both in terms of clinical manifestations and molecular profiles. It is increasingly recognized that AD is a systemic rather than a local disease and should be assessed in the context of whole-body pathophysiology. Here we show, via integrated RNA-sequencing of skin tissue and peripheral blood mononuclear cell (PBMC) samples along with clinical data from 115 AD patients and 14 matched healthy controls, that specific clinical presentations associate with matching differential molecular signatures. We establish a regression model based on transcriptome modules identified in weighted gene co-expression network analysis to extract molecular features associated with detailed clinical phenotypes of AD. The two main, qualitatively differential skin manifestations of AD, erythema and papulation are distinguished by differential immunological signatures. We further apply the regression model to a longitudinal dataset of 30 AD patients for personalized monitoring, highlighting patient heterogeneity in disease trajectories. The longitudinal features of blood tests and PBMC transcriptome modules identify three patient clusters which are aligned with clinical severity and reflect treatment history. Our approach thus serves as a framework for effective clinical investigation to gain a holistic view on the pathophysiology of complex human diseases.

Atopic dermatitis (AD) is one of the most common chronic inflammatory skin diseases worldwide and is characterized by a highly heterogeneous clinical phenotype[1,2]. Causal factors, disease course and underlying immunological pathways of AD vary greatly among patients, making clinical management tremendously complicated[3,4]. In spite of growing therapeutic options with a wave of development of novel targeted drugs such as an anti-IL-4Rα antibody[5] and an anti-IL-31Rα antibody[6], there is no consensus concerning therapeutic decisions for

individual patients[7]. In order to provide optimal treatment for each patient with maximum cost-effectiveness, there is an urgent need to characterize patients in terms of endotypes that are potentially linked with disease course.

Although recent advances in biomedical technologies have enabled us to acquire an enormous amount of patient omics data including genome data, capturing fundamental endophenotypes of individual patients is still challenging. In the past decades, multiple

attempts were made to uncover the biological features of skin tissues or peripheral blood mononuclear cells (PBMC) from AD patients using transcriptomic and proteomic approaches. Those studies have revealed important roles of Th2 or Th17 pathways both in skin and in PBMC along with altered skin barrier function in AD pathology[8–10], and some of them further demonstrated how these pathways can be targeted by systemic treatment with immunosuppressants[11], anti-IL-4Rα antibody[12,13] and oral JAK inhibitors[14]. However, these observations in either skin tissue or blood only focus on alterations in a specific part of the body that could reflect just one aspect of a highly complex pathology. It is widely recognized that complex diseases should be assessed in the context of whole-body level biology since organs are communicating with each other[15–17]. Projects such as GTEx[18] and Human Cell Atlas[19] can be utilized for per-tissue/cell type characterization of human biology, as well as characterization of inter-tissue communications ("crosstalk"). Skin disorders including AD, which is now recognized as a systemic disease[20,21], need special attention to such crosstalk between the originally damaged organ and the circulatory system[22,23]. The importance of considering cross-tissue interactions in skin immunological regulation is also supported by the evidence of concurrent biological alterations in both skin tissue and blood after systemic treatment in AD[12,24,25] or in HIV infection, which frequently causes cutaneous malignancies or inflammation[26].

Other essential factors in AD pathology include the heterogeneous disease trajectories as characterized by repeated exacerbation and remission, with different cycles by patients. Correspondingly, most patients have their own medication history over time, based on their incidence of exacerbations. Accounting for such heterogeneity in disease trajectory has been extremely challenging in previous omics-based studies of AD.

In this study, we carry out cross-sectional analysis and longitudinal analysis with observational datasets, aiming to capture biological signatures in the context of clinical profiles in the Japanese AD population. For the cross-sectional analysis, we analyze RNA-seq data of both skin and PBMC from AD patients and healthy controls and link them to clinical data. Via building regression models incorporating both skin and PBMC transcriptome data that are preprocessed into interpretable transcriptome modules, we establish factors that contribute to clinical presentations across patients. For the longitudinal analysis, we apply the transcriptome modules along with the regression models established in the analysis of cross-sectional dataset to a time series dataset to monitor personalized disease courses and to examine inter-patient heterogeneity in longitudinal features. These multifaceted analyses of cross-tissue, cross-sectional and longitudinal transcriptomes highlight the close association between phenotypes and endotypes in AD. Our approach serves as a framework for effective clinical investigation of heterogeneous and complex human diseases.

## Results
### Characterization of participants
A schematic presentation of the process of filtering samples and patients for each analysis was shown in Fig. S1. For cross-sectional analysis, 188 AD patients and 45 healthy controls were extracted from the overall sample collection according to the criteria defined in the section *Study design* in "Methods". RNA-seq data from samples either with low read count (total read count <5 million) or with a strong batch effect attributable to inadequate sample processing were excluded. Consequently, 151 AD patients and 19 healthy controls that met the criteria for RNA-seq data were extracted. Patients were further filtered by gene expression intensity of pilosebaceous unit-related genes in skin samples (Fig. S2), resulting in 115 AD patients and 14 healthy controls as eligible samples (one sample per patient) for regression analysis using all of skin, PBMC and blood tests (Fig. 1). Frequency distribution of the AD patients by disease severity is shown in Fig. S3. All the samples (315 skin samples and 235 PBMC samples from both AD patients and healthy controls) that were assured for RNA-seq data quality by itself were included for transcriptome modules identification to increase power. Of these participants, 27.1% (30 AD patients and 5 healthy controls) were female. Sex (biological attribute) of the participants was determined based on self-reporting. Their mean age was 41.3 years (AD: 40.5, healthy: 47.3, range, 21–70 years).

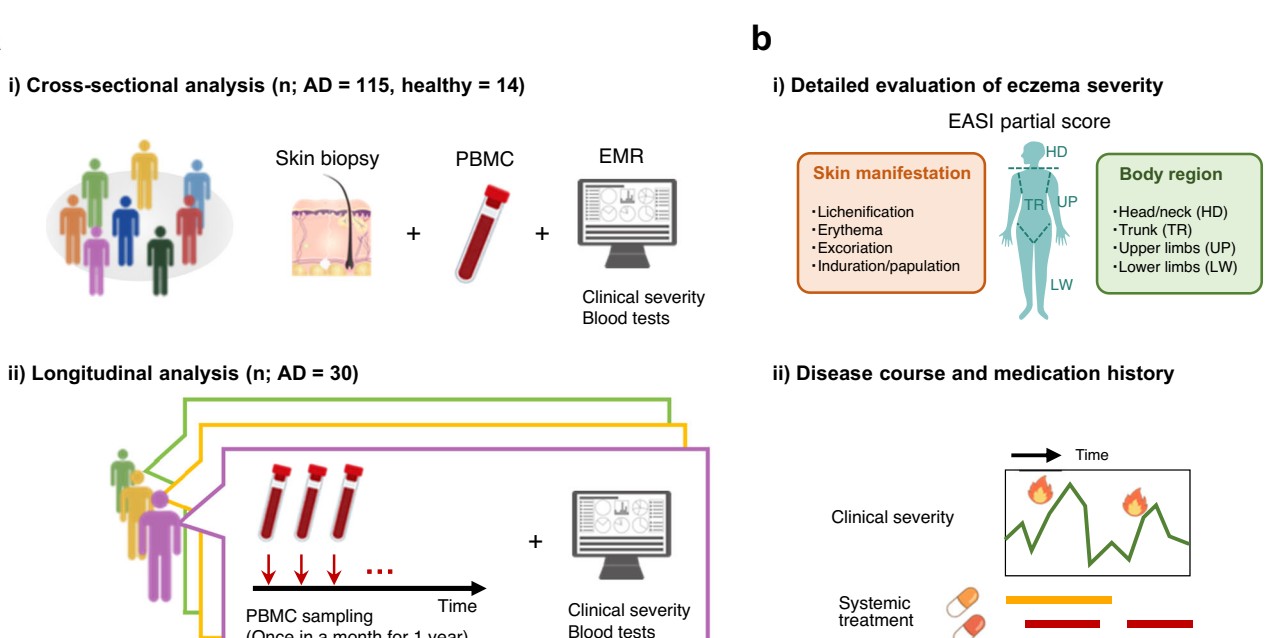

**Fig. 1 | Summary of study design. a** This study consists of two parts, i) a cross-sectional part (n; Atopic dermatitis: AD = 115, healthy = 14) and ii) a longitudinal part (n; AD = 30) to elucidate endotypes that are associated with phenotypes in AD. **b** We focused of two classes of disease phenotypes highlighted by clinical data; i) skin manifestation and ii) longitudinal disease course along with medication history, that were examined in association with endotypes in cross-sectional and longitudinal analysis, respectively. EMR electronic medical records, EASI Eczema Area and Severity Index.

For longitudinal analysis, time series dataset consisting of PBMC transcriptome, laboratory blood tests and clinical severity score from 30 AD patients on monthly basis up to a year (total 360 time points) were extracted, and after quality control, 280 data were considered as eligible and used for longitudinal analysis. Of these AD patients, 7 patients (23.3%) were female, and 30 patients (100%) and 17 patients (56.7%) overlapped with the cross-sectional population for PBMC only and PBMC + skin analysis, respectively.

For meta-analysis of clinical severity scores, we used a total of 1424 data points obtained during the period of November 2016 to July 2021 from the 151 AD patients who were included in cross-sectional and/or longitudinal analysis. The AD patients in this observational study were basically under treatment with topical steroids and emollients as directed by dermatologists, except for the 5 patients who refrained from using topical steroids for some reason. Their history of systemic treatment was categorized as follows: intermittent use of oral steroids, intermittent use of immunosuppressant, antiallergic agents with continuous use (a total of more than 120 days/year), antiallergic agents with occasional use (a total of fewer than 120 days/year), and no use of these agents. Drugs used for systemic treatment in the overall AD population in this study are listed in the Table S2. Characteristic information of the participants is summarized in Table S3.

## Compositional analysis of clinical scores highlighted two distinct skin manifestations

The extent and severity of atopic dermatitis were measured using the Eczema Area and Severity Index (EASI)[27]. In this scoring system, severity is determined by grading the key signs of eczema (i.e. erythema, induration/papulation, excoriation, and lichenification) over the four anatomic divisions of the body (i.e. the head and neck, the trunk, the upper extremities, and the lower extremities) separately. The average severity of each sign in each of the four body regions was assigned a score of 0–3 (none, mild, moderate, and severe, respectively).

To capture the relationship between individual components of eczema severity, we performed multidimensional scaling (MDS) which is a visual representation of distances between sets of objects[28] on the collection of partial scores across patients (Fig. 2a). Two major clusters were found in the aspect of key signs of eczema (Fig. S4); one consisted of erythema and lichenification and the other consisted of induration/papulation and excoriation. This suggested that erythema and induration/papulation constitute two distinct skin manifestations, apt to be accompanied by lichenification and excoriation, respectively, as signs of progression or chronicity. From a regional perspective, the configuration of the scores for the four body regions was all in the same order in the MDS plot, i.e. from left to right are the lower extremities, the upper extremities, the trunk to the head and neck, leaving the head and neck distant from the other three regions. This finding is consistent with the recent view that head and neck erythema is a prominent form of AD[29,30].

Based on these findings, we defined two distinct phenotypes in AD, an erythema form and a papulation form, using the summation of either erythema or papulation scores in all the body regions except for the head and neck, respectively. Meanwhile, we defined the general severity of AD as the summation of all the scores, i.e. EASI (total) as is conventionally used.

In order to pathologically characterize skin types of both erythema and papulation in AD, we conducted immunohistochemistry of lesional skin from the six erythema-skewed and the six papulation-skewed patients (Figs. S5–7, Fig. 2b, c). Figure 2b, c shows clinical and histological images of the representative patients who have a score composition that is highly skewed to either erythema or induration/papulation (partial score for the left patient; erythema = 9.6, papulation = 4.8, the right patient; erythema = 4.3, papulation = 8.6). Histological analysis revealed shared and differential characteristics in the skin tissue between the erythema- and papulation-skewed AD patients. In both skin samples, intense infiltration of immune cells including CD4+ T cell (Fig. 2c), macrophage (CD206+), myeloid dendritic cell (CD11c+, DC-LAMP+) and Langerhans cell (CD1a+), along with epidermal hyperplasia and diminished epidermal barrier (as observed by filaggrin expression) were commonly observed (Fig. S6). However, the patterns for immune cell infiltration appeared to be different between erythema and papulation; the skin sample from the erythema-skewed patient were characterized by diffuse infiltration of immune cells in dermis, accompanied by epidermal lymphocytic infiltration. On the other hand, the skin sample from the papulation-skewed patient was characterized by nodular infiltration of immune cells in dermis suggestive of geometrical heterogeneity over the lesion, as well as prominent hyperkeratosis. Those observations were largely reproduced in other five erythema-skewed patients and five papulation-skewed patients, respectively (Fig. S5). Neutrophil (myeloperoxidadse: MPO+) infiltration were substantially observed in the skin sample from the erythema-skewed patients but not in the skin sample from the papulation-skewed patients.

## Transcriptional characteristics of skin tissue and PBMC typically found in AD

To identify transcriptome signatures enriched in AD patients, we firstly conducted differential gene expression analysis on RNA-seq data of skin and PBMC specimens. Accordingly, 272 and 33 differentially expressed genes for skin and PBMC, respectively, were identified (|log2 fold change (log2FC)| ≥ 2 and false discovery rate (FDR) < 0.01 for skin and |log2FC| ≥ 1 and FDR < 0.05 for PBMC, Fig. 3a). Gene ontology (GO) terms enriched in skin of AD patients included antimicrobial peptides, chemokine and interleukin signaling genes and epidermal differentiation/keratinization, which is largely consistent with previous reports[10,31]. GO terms enriched in PBMC of AD patients included neutrophil degranulation and immune system (Fig. 3b).

## Inference in ligand-receptor coupling suggests augmented skin-PBMC crosstalk in AD patients

The increased expression of inflammation-related genes in both skin and PBMC suggested that inflammation induced in skin tissue in turn triggered inflammatory responses in PBMC, or vice versa in some cases, presumably through secretion of soluble factors that can act on cells in the circulatory system[22]. In order to illuminate such potential crosstalk between skin tissue and PBMC, we integrated RNA-seq data from both sources and quantified ligand-receptor couplings that are particularly engaged in inflammatory signaling[32].

We defined active cytokine–receptor pairs as having concurrent expression of a cytokine gene and its matching receptor gene at a level of cytokine gene > 0.5 and receptor gene > 0 in value of variance stabilizing transformation (vst) applied to the expression values that were followed by normalization across the population. A total of 210 pairs of inflammatory cytokine and receptor genes were assessed in the skin and PBMC of each AD patient and healthy control. The active cytokine–receptor pairs were enumerated according to classes defined by the combination of a sender organ that expressed a cytokine gene and a receiver organ that expressed a receptor gene (Fig. 4a; "Methods"). The total number of active cytokine–receptor pairs was significantly higher in AD patients than in healthy controls (mean = 50.9 vs 29.6; $p = 1.0E{-}3$).

Among these, the number of connections from skin to skin and the number of connections from skin to PBMC were significantly increased in AD patients compared to healthy controls (mean = 24.6 vs 10.9 and 17.3 vs 10.6; $p = 8.3E{-}5$ and $p = 2.8E{-}3$, respectively), while the number of connections from PBMC to either of skin or PBMC was not significantly different between AD patients and healthy controls

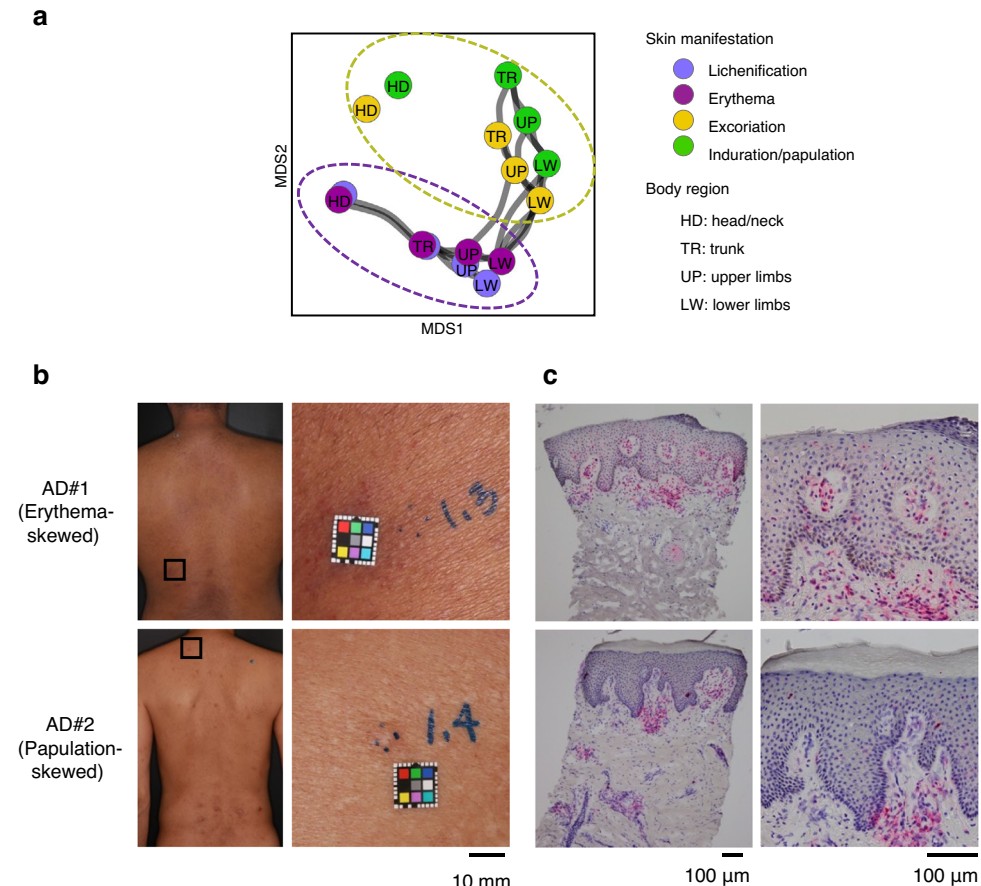

**Fig. 2 | Compositional analysis of clinical scores highlighted two distinct skin manifestations in AD. a** Separation pattern by multidimensional scaling (MDS) on individual components of EASI across AD patients. Components that are correlated with each other (*Pearson r* > 0.40) were connected with gray lines. Two major clusters were identified in the aspect of key signs of eczema, among which erythema and induration/papulation are two primary skin manifestations that bear the distinction. Clinical pictures (**b**) and immuno-histochemistry of skin tissue for CD4 (**c**, target protein was stained in red) in two representative patients who have a score composition that are skewed to either of erythema (upper) or induration/papulation (lower). Upper: a 51-year-old male patient who has erythema-skewed EASI composition (total = 19.6, erythema = 5.2, papulation = 3.4). Lower: a 50-year-old male patient who has papulation-skewed EASI composition (total = 21.0, erythema = 3.0, papulation = 8.4). One slide per patient was stained for one marker protein in histological analysis. Assays for other markers in the same patient samples are shown in Supplementary Figs. 6 and 7.

(Fig. 4b). Stratified analysis on skin−PBMC interaction revealed progressive augmentation of the number of links in severe AD compared to moderate and mild AD, suggesting that systemic inflammation is more evidently involved in severe AD, although there was no statistical difference among three groups (Fig. S8). There were moderate correlations between the total number of cytokine−receptor connections and either EASI ($r = 0.32$; $p = 2.5E-4$) or serum TARC ($r = 0.35$; $p = 6.6E-5$, Fig. 4c).

The most frequently observed pairs in AD were *CCL22-CCR4/* and *CCL17-CCR4* in skin, while in healthy controls they were *IL37* (skin) - *IL18R1/IL18RAP* (PBMC) and *IL34* (skin) - *CSF1R* (skin). The top two frequently observed pairs involving PBMC in AD were *CCL18* (skin) - *CCR8* (PBMC) and *IL20* (skin) – *IL20RB* (PBMC) (Table S4). Cell types responsible for expression of these cytokine/receptor genes were estimated by referring to publicly available datasets that are suitable for analyzing cell type expression[33,34]. The most frequently appearing cell types in AD were T cells and vascular endothelial cells (VEC) as cytokine-expressing cells, and myeloid cells and T cells as receptor-expressing cells, all of which were found in the skin. The most highly involved cell type in PBMC was the monocyte, for both cytokine and receptor expression (Table S5). Collectively, the indication of enhanced ligand-receptor coupling involving both skin and the circulatory system in AD patients suggested the need for a system-level investigation into AD pathology.

## Identification and characterization of transcriptional modules associated with AD

To illuminate the heterogeneity in the biological signature across AD patients, expression levels of not only DEGs between AD patients and healthy controls but also the extended range of gene sets that have potential association with AD pathology should be analyzed. Weighted gene co-expression network analysis (WGCNA) is a powerful technique to depict functional subsystems by highlighting biologically relevant transcripts with reduced dimensionality across a population[35]. We applied WGCNA to our entire expression dataset, including AD patients and healthy controls for skin and PBMC, respectively to identify AD-related transcriptional modules. This procedure identified 21 skin transcriptional modules (sModus) and 15 PBMC transcriptional modules (pModus), each comprising 51–774 genes (mean; 258.7 for skin, 191.8 for PBMC) that behave synchronously in a tissue, suggesting their biological relevance to each other (Fig. 5a).

As expected, genes in each module exhibited substantial cell type specificity in their expression as confirmed by referring to the publicly available dataset of either single-cell RNA-seq (scRNA-seq) (for skin) or sorted cell RNA-seq (for PBMC). Figures 5b and d show the size of the first principal component (PC1) value per cell type obtained by applying principal component analysis (PCA) on gene expression data of cell types for each gene module (i.e. matrix with $m$ columns of gene and $n$ rows of cell types, where $m$ is the number of

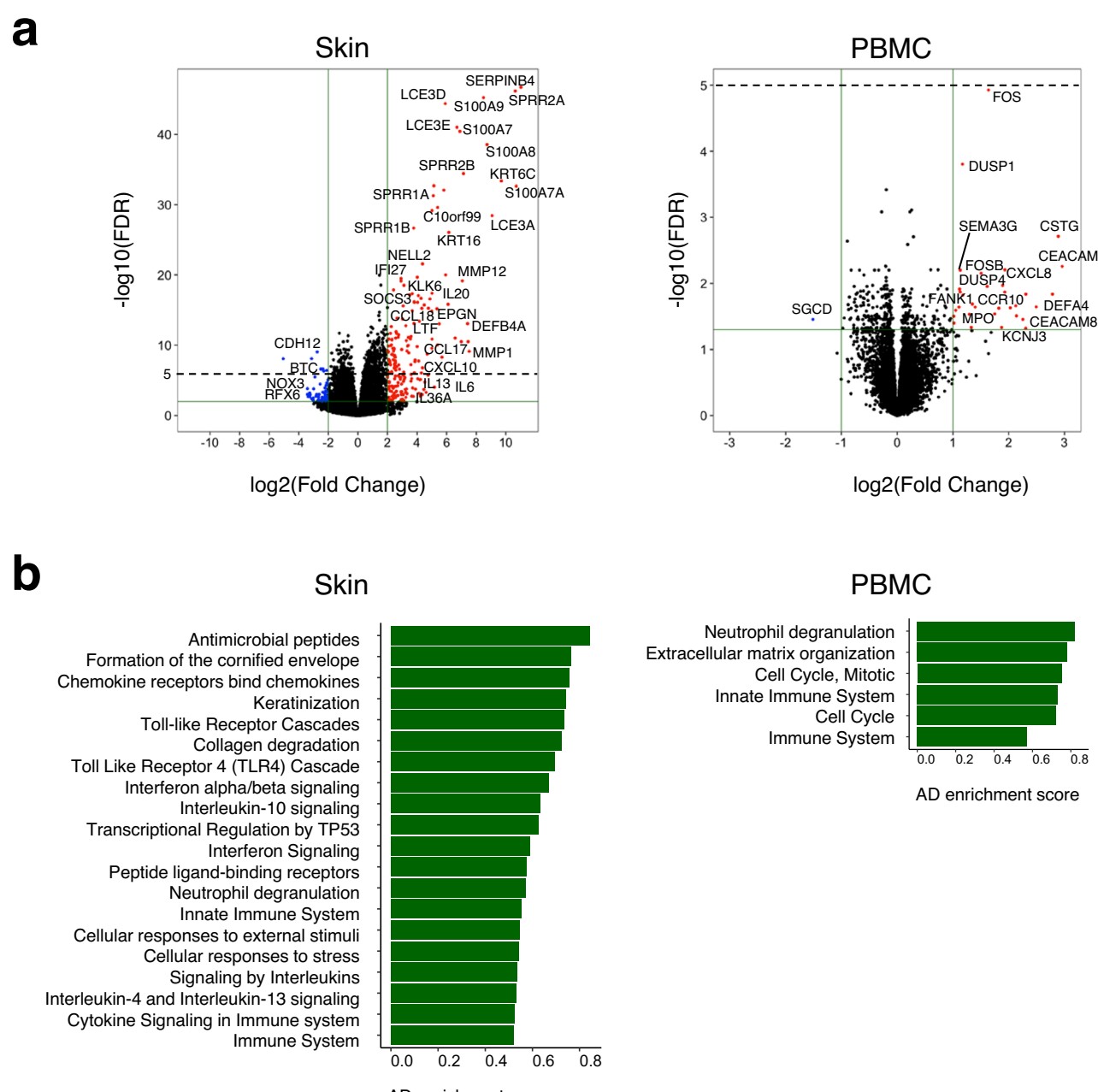

**Fig. 3 | General transcriptional characteristics of skin and PBMC in AD.**
**a** Volcano plot with significantly differentially expressed genes (|log2 fold change (log2FC)| ≥ 2 and false discovery rate (FDR) < 0.01 for skin and |log2FC| ≥1 and FDR < 0.05 for PBMC) highlighted in red (up-regulated in AD) and blue (down-regulated in AD) compared to healthy controls (n; AD = 115, healthy = 14). **b** Gene ontology (GO) terms enriched in differentially expressed genes in AD (FDR < 0.1 and Enrichment score > 0.5). Enrichment score was obtained based on the size of a given gene set in GO terms (see "Methods"). Source data are provided as a Source Data file.

genes assigned to a given module). See Fig. S10 and Supplementary note for further characterization of the gene modules. Relationships among the top 30 genes of the first principal component (PC1) from each module were visualized on the basis of gene-gene networks using thresholding of eigengene-based connectivity > 0.65 (Fig. 5c, e). This analysis revealed several notable signaling compartments in each tissue; compartments of acquired immune regulation (cytokine signaling), innate immune regulation (interferon signaling) and compartments of keratinization/formation of cornified envelope, in skin tissue. Additionally, three modules were found to be representing skin appendages; sebaceous gland (sModu01, GO: fatty acid metabolism) and sweat gland (sModu03 and sModu19, GO: ion

channel transport and developmental biology, respectively). The intensity of these modules was not relevant to dermatitis, and was strongly biased by sampling regions. Therefore, we considered these two modules as noises, and excluded from the following analysis. Another potential representation of skin appendage, although it is not evident as much as above mentioned three modules, is a neuroreceptor signature by sModu02 which include *KCNH4*, *CACNA1A* and *ASIC2*, genes coding ion channel subunits with suggested association with sensory neuron in human skin[36].

To obtain personalized profiles based on the transcriptional modules, scores for each module and each patient were defined. Since identified modules consist of co-expressing genes, expression

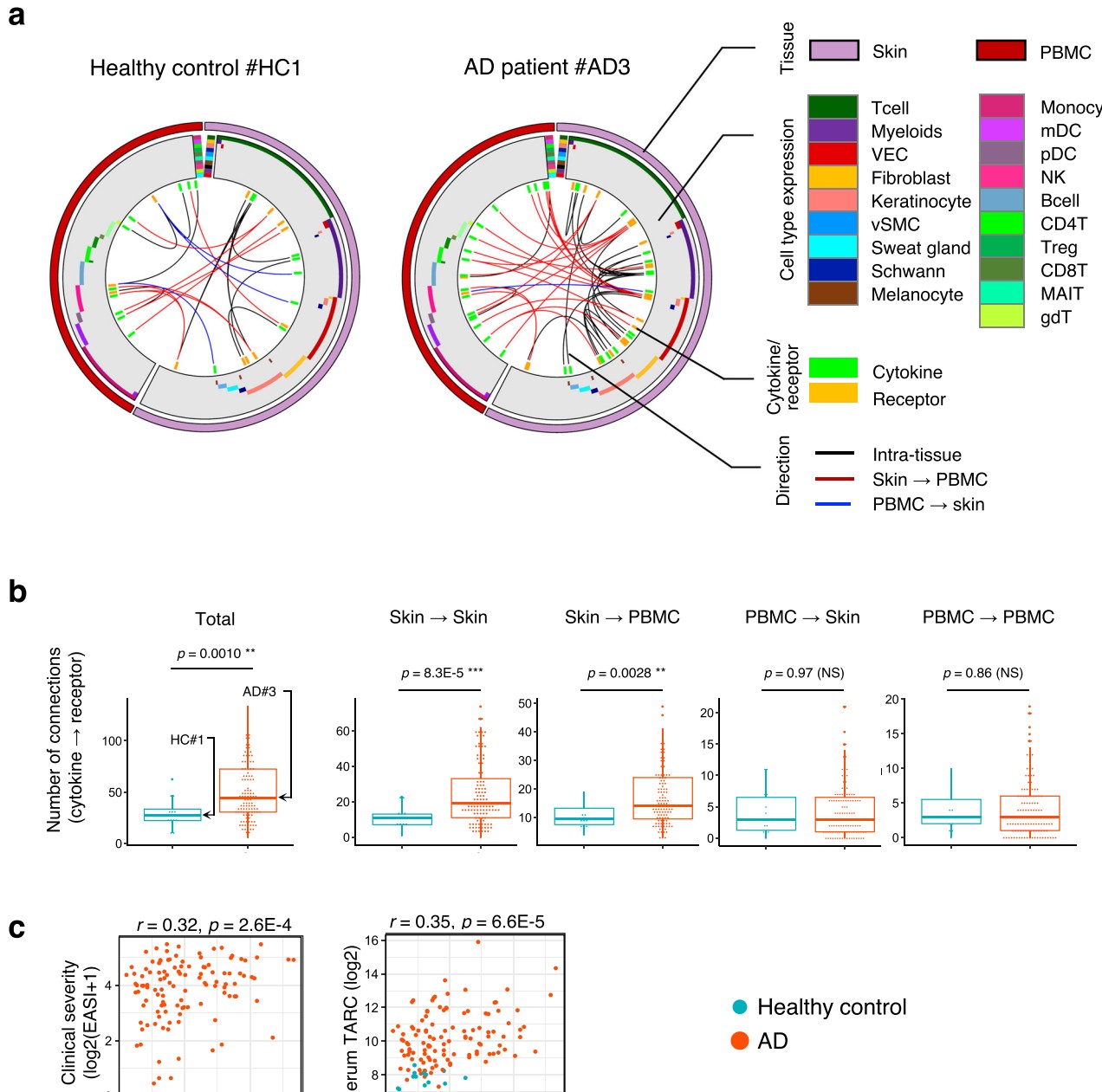

**Fig. 4 | Inference in ligand-receptor coupling suggests augmented skin-PBMC crosstalk in AD patients. a** Connection map of cytokine–receptor coupling across skin and PBMC in a representative healthy control (left) and AD patient (right). Genes that code cytokines and receptors are aligned along the perimeter of the circles. From the outer layer to the center is the tissue expressing the genes (either skin or PBMC), inferred cell specificity, classification of cytokine or receptor, and the connections between cytokines and its matching receptors. The connections were indicated in different colors according to the classification of direction, i.e. in which tissue the cytokines are produced and on which tissue they act. VEC: vascular endothelia cell, vSMC: vascular smooth muscle cell. **b** Number of active connections between cytokines and receptors. Connections were enumerated according to 4 classes defined by a sender organ and a receiver organ. Boxplots show median and first and third quartiles, whiskers extending to the highest and lowest values no further than 1.5*interquartile range. Brunner-Munzel rank test, two-sided, **p < 0.01, NS: not significant. **c** Pearson correlation between number of active connections and clinical index, two-sided. N; Atopic dermatitis: AD = 115, healthy = 14 (biologically independent samples). Source data are provided as a Source Data file.

patterns in each module became simple enough to be handled linearly, as verified by substantially high value of explanatory capability of PC1 (40–60%) when PCA was applied on gene expression data of patients for each gene module. Therefore, we used the PC1 values followed by standardization across patients as the index of intensity of gene expression of transcriptome modules in each patient (Fig. S9).

## Regression analysis reveals differential patterns of modular involvement in erythematous and papular skin manifestations in AD

We next investigated how the AD phenotypes can be represented by transcriptome modules from both skin and PBMC, as well as by laboratory tests (Table S6) obtained at the same visits. Given the relatively large number of variables to the sample size, we built

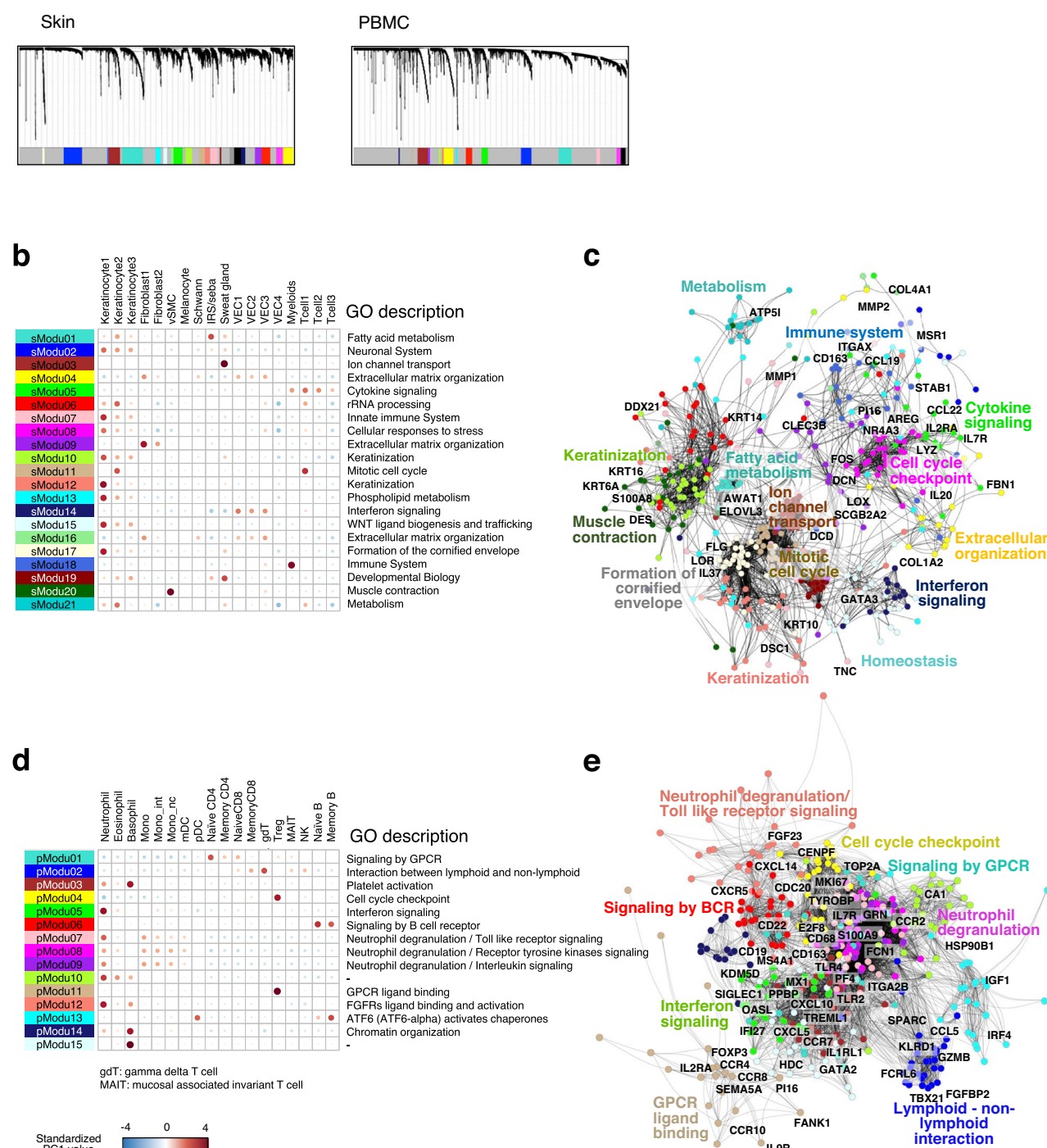

**Fig. 5 | Identification and characterization of transcriptional modules from skin/PBMC RNA-seq data. a** Cluster dendrograms of transcripts produced by implementation of WGCNA. Color indicates separation of transcriptional module. Cell type expression and GO enrichment in skin tissue (**b**) and PBMC (**d**) analyzed by referring public database. Visualization of gene-gene networks in PC1 top 30 genes from each transcriptome module in skin (**c**) and PBMC (**e**). Genes that have eigengene-based connectivity > 0.65 were connected with lines. sModu skin transcriptome module, pModu PBMC transcriptome module, vSMC vascular smooth muscle cell, IRS/seba inner root sheth/sebaceous gland, VEC vascular endothelia cell.

regression models using elastic net, an algorithm for regularized regression and variable selection that is applicable to high dimensional data with multicollinearity[37].

To confirm that regularized regression is superior to linear model in building regression models on our complex dataset consisting of both skin and PBMC transcriptome, we compared the performance of the linear model and elastic net (Fig. 6a). We found

that adj $R^2$ for the test dataset is higher in elastic net model compared to linear model (adj $R^2$ (training) = 0.65, $R^2$ (test) = 0.02 for linear model vs adj $R^2$ (training) = 0.64, $R^2$ (test) = 0.43 for elastic net model, when all the variables are used), verifying the advantage of elastic net in our data. Although addition of transcriptional modules did not improve the overall model performance drastically, several gene modules that were selected as the predictor variables for the AD

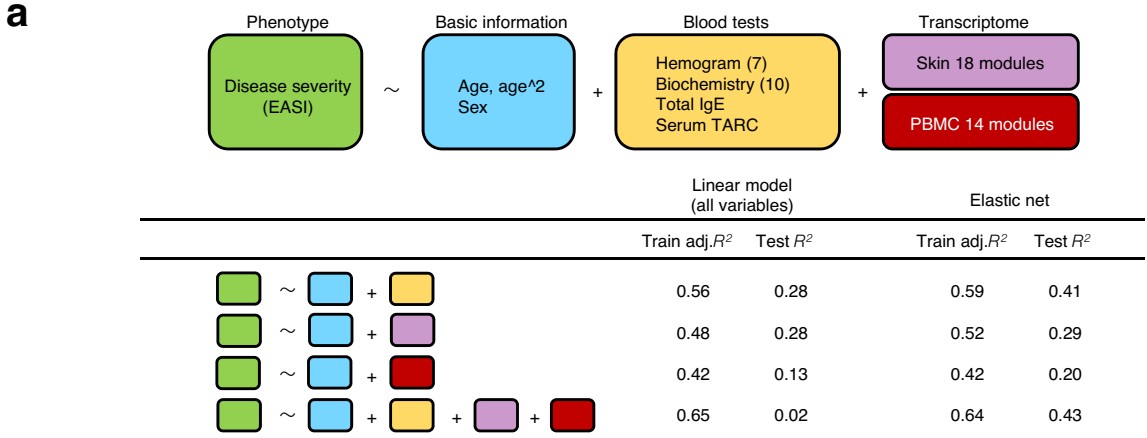

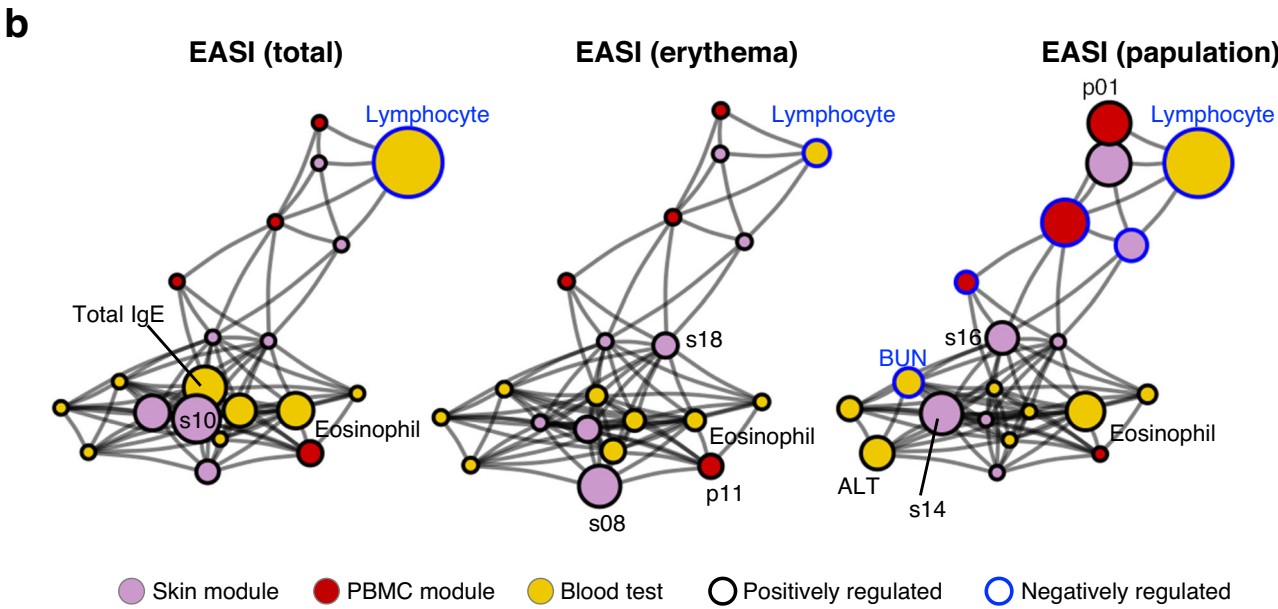

**Fig. 6 | Regression analysis revealed differential patterns of modular involvement in erythema and papulation skin manifestation in AD. a** Regression models for the prediction of clinical phenotypes. Adjustment was made for $R^2$ in training set with the number of prediction variables. **b** Predicted dysregulated networks of blood tests and skin/PBMC transcriptome modules contributing to distinct phenotypes. Node size and node frame color represent size and the sign of coefficients for each variable predicted by elastic net regression. sXX skin transcriptome module XX, pXX PBMC transcriptome module XX, ALT alanine transaminase, BUN blood urea nitrogen.

phenotypes provided insights into transcriptional regulation involving pathology.

We also built elastic net regression models to predict EASI (erythema) and EASI (papulation) which made a major distinction in skin manifestations as described above. The model performance ($R^2$) and the set of significant features ($p < 0.05$) in each model as well as its biological characteristics are summarized in Table 1.

Both in erythema and in papulation skin manifestation, a decreased lymphocyte ratio in blood, an indication of increased proportion of myeloid cells (a populational summation of monocytes and neutrophils) and an increased eosinophil ratio in blood were found to be associated with symptoms. Erythema was characterized by a bolstered signature of immediate early genes (*NR4A1, FOSL1, FOSB, ATF3, NR4A2*) and immune system (*CD163, C1QB, C1QC, THY1, MS4A7*) that are inferentially expressed mainly in keratinocytes and myeloid cells, respectively, in skin tissue, along with Treg specific genes (*CCR4, CNTNAP1, DUSP4, LMNA, PI16*) in PBMC. In contrast, papulation was characterized by decreased B cell signature (*FCRL1, MS4A1,*

*PAX5, CD22, LINC00926*) and increased naive CD4 signature (*NELL2, LRRN3, OBSCN, CCR7, GRASP1*) in PBMC along with enhanced signature of interferon signaling (*MMP12, CCL18, IFI27, TYMP, COL6A6*) and extracellular matrix (*PI15, GREM1, COL4A1, TNFAIP6, NNMT*), suggestive of altered activity in VEC and fibroblast in skin tissue. We confirmed by subanalysis that these results were not biased by the potential influence of the treatment difference among patients (Table S7, Supplementary note). Dysregulated module networks contributing to distinct phenotypes were predicted based on the coefficient of each variable (Fig. 6b). These results suggest that pathologies underlying erythema and papulation are substantially different on a molecular basis.

**Personalized monitoring of trajectory of disease severity and molecular signatures**

One of the most essential features of AD is that patients follow a disease course complicated by exacerbations and remissions throughout the years, thereby patients take individual treatment steps based on their

**Table 1 | Prediction variables extracted in regression models**

| Objective variables | $R^2$ | Predictors | Coefficient | P-value | Tissue | PC1 top 5 genes | Cell type specificity |
|---|---|---|---|---|---|---|---|
| EASI (total) | Training 0.61 Test 0.43 | Lymphocyte | −0.40 | 8.80E−04 | Blood | – | Lymphocyte |
| | | Total IgE | 0.25 | 0.042 | Blood | – | – |
| | | Eosinophil | 0.21 | 0.043 | Blood | – | Eosinophil |
| | | sModu10 | 0.27 | 0.081 | Skin | S100A8, S100A9, KRT6C, SERPINB4, S100A7 | Keratinocyte |
| EASI (erythema) | Training 0.63 Test 0.51 | sModu08 | 0.22 | 2.60E−03 | Skin | NR4A1, FOSL1, FOSB, ATF3, NR4A2 | Keratinocyte |
| | | Lymphocyte | −0.14 | 0.041 | Skin | – | Lymphocyte |
| | | sModu18 | 0.13 | 0.076 | Blood | CD163, C1QB, C1QC, THY1, MS4A7 | Myeloids |
| | | pModu11 | 0.13 | 0.077 | Skin | CCR4, CNTNAP1, DUSP4, LMNA, PI16 | Treg |
| | | Eosinophil | 0.11 | 0.099 | Blood | – | Eosinophil |
| EASI (papulation) | Training 0.54 Test 0.33 | Lymphocyte | −0.39 | 7.50E−04 | Blood | – | Lymphocyte |
| | | pModu06 | −0.27 | 0.0029 | Blood | FCRL1, MS4A1, PAX5, CD22, LINC00926 | B cell |
| | | Eosinophil | 0.22 | 0.0091 | Blood | – | Eosinophil |
| | | pModu01 | 0.24 | 0.015 | Blood | NELL2, LRRN3, OBSCN, CCR7, GRASP1 | Naive CD4 |
| | | ALT | 0.19 | 0.017 | Blood | – | – |
| | | BUN | −0.17 | 0.03 | Blood | – | – |
| | | sModu14 | 0.24 | 0.031 | Skin | MMP12, CCL18, IFI27, TYMP, COL6A6 | VEC |
| | | sModu16 | 0.18 | 0.073 | Skin | PI15, GREM1, COL4A1, TNFAIP6, NNMT | Fibroblast |

Elastic net regression was applied to data including basic information, blood test, skin transcriptome modules and PBMC transcriptome modules. Adjustment was made for $R^2$ in training set with the number of prediction variables. $N = 129$ (Atopic dermatitis: AD = 115, healthy = 14).
sModu skin transcriptome module, pModu PBMC transcriptome module, ALT alanine transaminase, BUN blood urea nitrogen, VEC vascular endothelial cells.

condition at a given time[38,39]. To provide an overview including symptom changes and use of systemic treatment in individual patients, we conducted monthly monitoring of PBMC transcriptomes, laboratory tests and severity scores for 30 AD patients for up to a year. We leveraged transcriptomic modules generated in the cross-sectional patient dataset and profiled the dynamics of transcriptomic features as well as blood tests that were lastly analyzed in association with disease severity.

We first tested the performance of elastic net regression model trained with cross-sectional dataset when the model was applied to the longitudinal dataset (Fig. 7a). Prediction performance for EASI (total) was higher in a model using all of basic information (age, age^2, sex), laboratory tests and PBMC transcriptome compared to a model using only basic information and laboratory tests ($R^2$: 0.15 vs −0.24).

Taking a closer look at individual trajectories of disease course, we found substantial variability in prediction accuracy among patients. Personalized disease trajectories in two representative patients are shown as examples in Fig. 7b. In the first example, the prediction seemed successful (Pearson $r = 0.81$; $p = 2.4E-3$), accurately capturing the disease flare (month 5). In contrast, prediction was unsuccessful in the second example as evident by Pearson $r = −0.44$ ($p = 0.20$). There was no significant difference in prediction accuracy of personalized trajectories of disease severity among patients regarding treatment classes (Fig. S11). We found that the time-course trajectory of the weights of TARC which was selected as the top predictor varibale in the elastic net model, strongly correlated with disease severity trajectory (Pearson $r = 0.88$; $p = 3.1E-4$) in the first example, but not in the second example (Pearson $r = −0.047$; $p = 0.91$) (Fig. S12). These observations suggest that the predominant features associated with disease course vary by patients, which could limit the performance of linear models assuming same feature weights across samples. Application of linear mixed model (LMM) on each analyte in the time series data also highlighted the varying random effects by patients (Fig. S13, Supplementary note).

**Close association between endotypic longitudinal features and phenotypic longitudinal features**

Given that another factor that accounts for the endotypes in individual AD patients is longitudinal variability itself[38], just as in other chronic inflammatory diseases[40], it is important to evaluate time series features in clinical severity and transcriptome modules. Seven types of time series features, i.e. mean, minimum, maximum, root mean square (RMS), mean absolute change (MAC), approximate entropy, and complexity-invariant distance (CID) were extracted from three categories of datasets, i.e. blood tests, PBMC transcriptome modules and clinical severity (i.e. EASI) in individual patients at a monthly interval over 1 year using the Python module Tsfresh[41] (Fig. 8a).

Hierarchical clustering of those 7 features of clinical severity in 30 AD patients showed two major clusters; one includes mean, maximum, minimum and RMS, the other includes MAC, CID and approximate entropy (Fig. S14). Therefore, we picked mean and MAC as representative values in two clusters, respectively, for demonstration of feature distribution among patients. Unsupervised k-means clustering on 30 AD patients based on time series features of PBMC transcriptome modules and blood tests, with number of clusters (= k) determined using silhouette criterion, identified three patient clusters (Fig. 8b). We applied PCA to this data of time series features to capture the patient distribution in a reduced dimension with the underlying structure that are differential across patients (Fig. 8c), and evaluated the intensity of the top PC1/PC2 contributing factors (Fig. 8d). Cluster 1 ($n = 2$) was characterized by stably high levels of pModu07 (GO: neutrophil degranulation/Toll-like receptor signal), pModu09 (GO: neutrophil degranulation/interleukin signaling) and neutrophil (complete blood counts-derived ratio: CBC) and a stably low level of lymphocyte (CBC), whereas Cluster 2 ($n = 7$) showed volatile trajectories of all of those terms throughout the observation period, as observed by high values of MAC with a medium level of mean. An unstably high white blood cell (WBC) count was also observed. Meanwhile, Cluster 3 ($n = 21$) was characterized by relatively low levels of all of those terms except for lymphocyte (CBC), which was relatively high in this patient cluster (Fig. 8d, Fig. S15).

Remarkably, those patient clusters were found to show clinical phenotypes associated with endotypic longitudinal features. Cluster 1 showed severe and stable symptoms, Cluster 2 showed severe and unstable symptoms, and Cluster 3 showed mild symptoms (Fig. 8e). Additionally, this patient grouping was found to be closely linked with prescription status of systemic treatment (Fig. 8f). Cluster 2, which

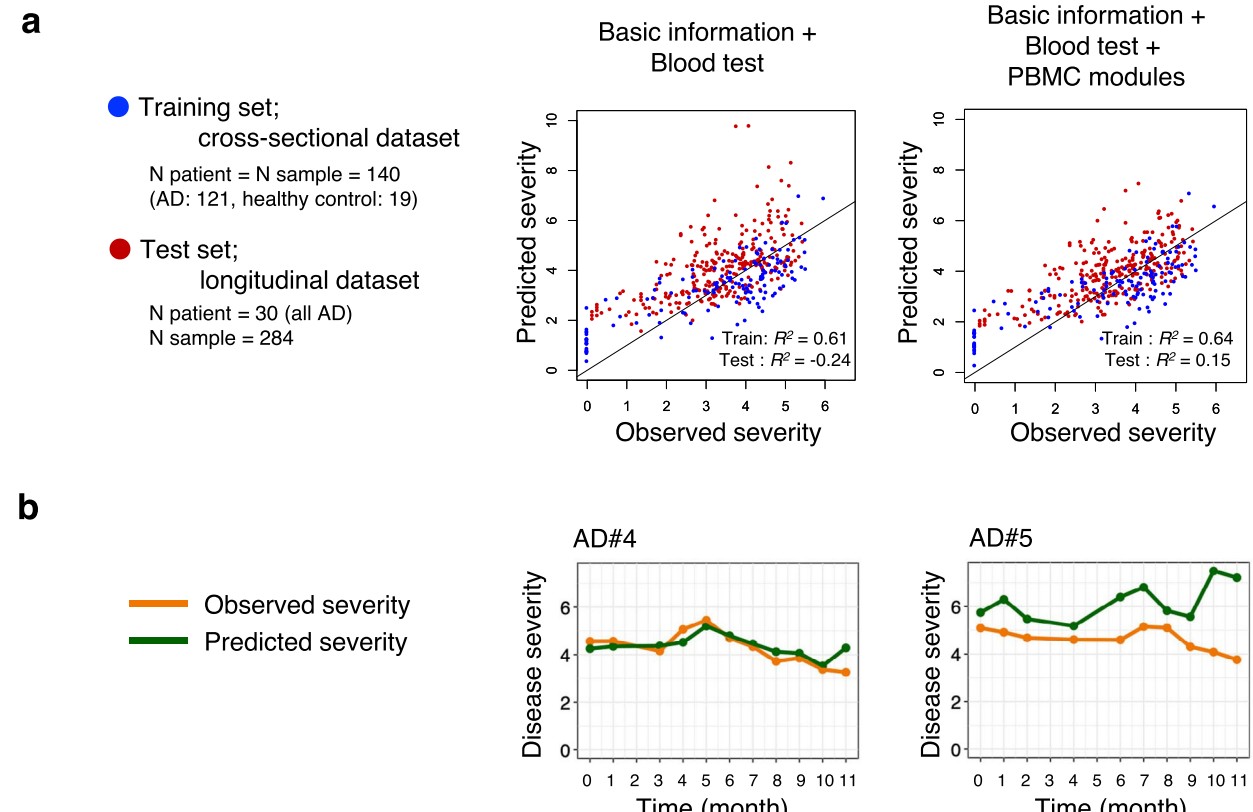

**Fig. 7 | Prediction performance of regression models on longitudinal dataset.**
**a** Performance of elastic net regression models to predict general disease severity (log2(EASI.total+1)). Models were trained with cross-sectional patient dataset and tested on longitudinal dataset. Adjustment was made for $R^2$ in training set with the number of prediction variables. **b** Trajectories of observed and predicted disease severity (log2(EASI.total+1)) in two representative patients both with successful prediction outcome (left, $r = 0.81$, $p = 2.4E{-}3$) and with unsuccessful prediction outcome (right, $r = -0.44$, $p = 0.20$) assessed by two-sided Pearson correlation. Source data are provided as a Source Data file.

manifested severe and unstable symptoms, highly overlapped with the patients who were under systemic therapy with an oral immunosuppressant (5/7 patient overlap), while Cluster 1 and Cluster 3 were mostly managing the disease either with antihistamines only or without systemic treatment.

The dynamics of EASI (total), top PC1/PC2 contributing factors, as well as the treatment periods in representative patients are shown in Fig. 8g. One possible logic for the observation of patient overlap between disease severity/stability and systemic treatment is that only severe patients are supposed to be candidates for systemic immunosuppressant therapy that leads to rapid symptom mitigation and global transcriptome alterations[13], but could cause a flare at the time of drug cessation. Patients treated with immunosuppressants in this study were all administered with the drug intermittently as instructed by their dermatologists, considering their symptom improvement or the risk of side effects. Accordingly, some patients experienced disease flare during washout periods. We thus note that systemic immunosuppressant therapy could partially contribute to instability of disease severity trajectory as well as other personal time series features.

## Discussion

With an increase of therapeutic options expected in the coming years, (1) understanding heterogeneity in disease phenotypes and endotypes and (2) patient stratification into subgroups based on phenotypes or endotypes, are the two urgent tasks for the development of personalized medicine in AD. Phenotypic heterogeneity among AD patients, which has been empirically recognized though not yet clearly defined, includes variability in skin manifestation and longitudinal disease course. In this study, we sought to elucidate endotypic heterogeneity

in association with these two aspects of phenotype, aiming at providing clinically significant and applicable insights in dermatology.

We profiled patients with transcriptome analysis on skin and blood biospecimens, each reflecting different aspects of disease state; skin for primary pathology at the site of ongoing or probable inflammation[10], and blood, a relatively homogeneous compartment, for systemic regulation of inflammation[42]. Although previous studies have reported patient stratification in AD based on single tissue data such as serum cytokine profiles[43], whole blood transcriptomes[44], or skin barrier profiles of comorbidity-stratified patient groups (with/without food allergy)[45], there are few reports on clinically significant endotypes regarding both skin and the circulatory system so far. He et al. demonstrated that patient groups defined on the basis of disease severity have differential molecular profiles in both non-lesional skin and serum[46]. Indeed, clinical manifestations in AD should be evaluated beyond the criterion of simple severity, given that several specific detailed signs of eczema have long been recognized in AD[27] including erythema and papulation, two distinct skin manifestations highlighted in our cross-sectional analysis. Exploring molecular involvement in such specific phenotypes using both skin and PBMC data should provide deeper insights into the unique characteristics of individual patients than in the case where the focus is on conventional general severity or just the presence of disease (AD versus healthy controls).

Our combinatorial approach of WGCNA and elastic net regression enabled us to efficiently and jointly analyze high dimensional datasets of skin and PBMC transcriptomes. Our finding on skin manifestation-dependent molecular profiles suggests that endotypes in AD (i.e. biological subtypes that were defined based on tissue

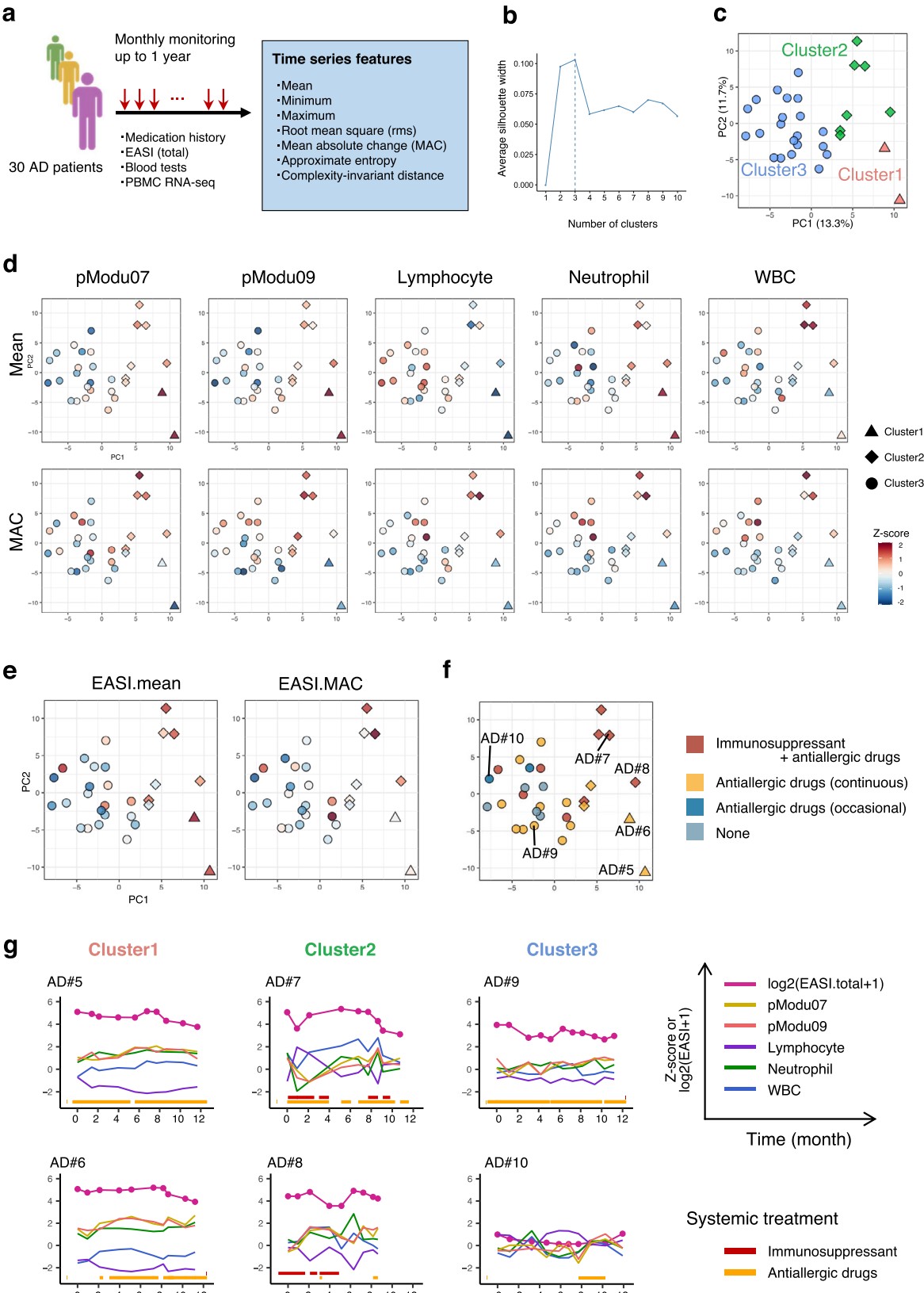

**Fig. 8 | Time series features of disease severity, clinical lab and PBMC transcriptome in each patient in association with history of systemic therapy.**
**a** Schematic of extraction of time series features in 30 AD patients. **b** Silhouette width plot for identifying the optimal number of patient clusters based on time series features. **c**–**f** PCA on 30 AD patients using time series features of blood tests and PBMC transcriptome modules. Color indicates patient clusters defined by k-means (**c**), the intensity of time series feature (upper; mean, lower; MAC) of 5 variables normalized among patients (**d**), time series features of clinical severity (**e**), and history of internal medication (**f**). **g** Dynamics of EASI (total), pModu07, pModu09, lymphocyte, neutrophil and WBC as well as period of internal medication in representative patients. MAC mean absolute change, pModu PBMC transcriptome module, WBC white blood cell. Source data are provided as a Source Data file.

transcriptome analysis in our study) are closely associated with the phenotypes of AD that were defined by visual evaluation of the skin. More fundamentally, this observation supports the assumption that the AD population comprises a variety of pathophysiological subtypes. Our report demonstrates association between endotypes and phenotypes with granularity beyond general clinical severity in AD.

Furthermore, we assessed heterogeneity in personal longitudinal features in PBMC transcriptome modules and blood tests in association with clinical severity. We identified three patient clusters based on longitudinal blood-derived signatures that were found to be closely linked with disease course and medication history. Our demonstration is the first step of patient stratification in the view of longitudinal features in AD, serving as a significant movement toward the grand challenge of personalized medicine.

There were also biological findings in the longitudinal analysis. Three top contributing factors for patient clustering was pModu07, pModu09, and neutrophil count, all of which was signatures reflecting innate immunity activity. This suggested that the dynamics of innate immunity may be the major force for instability in longitudinal disease course. As to the factors correlated with disease severity in individual patients, in addition to serum TARC, LDH and eosinophil counts, all of which are well-recognized biomarkers in AD[47], newly defined PBMC transcriptome modules including pModu01 (inferred cell specificity: naive CD4, PC1 top genes: *NELL2, LRRN3, OBSCN, CCR7* and *GRASP1*) and pModu04 (inferred cell specificity: Treg, PC1 top genes: *MKI67, RRM2, TOP2A, ASPM* and *MYBL2*) were identified as contributing factors in a personal disease course.

Although our study demonstrated integrative analysis of transcriptome data both from primarily diseased tissue and from circulatory system is advantageous for understanding patient endotypes, such assessment could not be applied in routine clinical examination especially in the longitudinal contexts, since acquiring biospecimen other than blood requires invasive sampling. Our next task is therefore, to identify representative biomarkers that can predict system-level pathology in individual patients only by routine clinical examination.

There are some limitations in this study. First, the clinical definition of skin phenotype manifestations is not totally objective. Scoring for severity of eczema was based on visual evaluation, which is strongly dependent on the expertise and experience of the dermatologists. The fact that most AD patients manifested multiple signs of eczema including erythema and papules simultaneously, with blurred boundaries, makes this issue even more of a problem. In the future, skin manifestations should be computationally and quantitatively evaluated, for example, through the abundance of hemoglobin or pigmentation in the skin, as has been investigated in some other skin disorders[48,49]. Second, our transcriptome data is from bulk RNA-seq which yields mixed signatures of different cell types in the tissue. Although we could infer cell type specificity for each molecular signature by deconvolution taking advantage of external scRNA-seq data, the resolution and accuracy is limited compared to the original scRNA-seq data[50–52]. Other limitations in our study includes limited sample size and population diversity, as is always the challenge in studies on complex human diseases. Above all, the AD patients in our cohort were enrolled in the single university hospital and can be potentially characterized by specific spectrum in disease severity. Studies with extended sample size and diversity may illuminate more profound heterogeneity in AD. Including non-lesional skin in the analysis would also serve this purpose since non-lesional skin could be a representation for congenital epidermal barrier function or immune regulation in pre-disease states. On the whole, our study highlighted inter- and intra- patient heterogeneity in AD, and demonstrated the promises of personalized AD treatment.

## Methods

### Study design

This study was approved by the Keio University School of Medicine Ethics Committee (Approval Number 20150325, 20160225, 20160131 and 20160377) and the RIKEN Ethics Committee (Approval Number H28-24) and conducted according to all relevant requirements from the Declaration of Helsinki. Written informed consent on sample collection, data acquisition and usage, and publication was obtained from all the participants. Participants received 5000 yen at one sampling of biospecimen for compensation for discomfort or inconvenience. Diagnosis of AD was made according to diagnostic criteria of Hanifin and Rajka[53].

We enrolled 196 Japanese AD patients who visited Keio University hospital and 46 healthy controls for skin and blood sampling study between December 2016 and February 2020 via information posters and documents. Pregnant or breast-feeding women, patients with episodes of lidocaine allergy, prilocaine allergy, or complications of bleeding disorders were excluded from recruitment. For cross-sectional analysis, we extracted eligible sample population based on the following criteria:(1) 20 years of age or older, (2) not being under systemic therapy with anti-IL-4Rα mAb nor JAK inhibitors, (3) having undergone biopsy from the back for skin samples. Accordingly, 188 AD patients and 45 healthy controls were extracted, and after data quality control as described in *RNA-seq and data processing* section as well as filtering with missing values in blood tests, 121 AD patients and 19 healthy controls were considered to be eligible for regression analysis on PBMC and blood tests, and 115 AD patients were considered to be eligible for regression analysis on all of skin, PBMC and blood tests.

For longitudinal analysis, samples from 30 AD patients who were enrolled in prospective observational study between December 2016 and September 2018 were analyzed. Time series dataset consisting of PBMC transcriptome, laboratory blood tests and clinical severity score from 30 AD patients on monthly basis up to a year (total 360 time points), were extracted. After data quality control, 280 data were considered to be eligible and used for analysis.

All the patients included in two analyses were treated according to the Japanese guideline for atopic dermatitis[54], such as emollients, topical corticosteroids and/or tacrolimus, oral antihistamines and immunosuppressants[29]. Note that the use of antihistamines was recommended as an adjuvant therapy to anti-inflammatory topical therapy to reduce itchiness in the treatment policy proposed by the Japanese guideline at the moment (i.e. 2016–2020)[54]. This policy was later modified to lower the grading of recommendation for the use of antihistamines in the revised guideline in 2021 in response to the increased recognition of uncertainty of its efficacy on relief of itchiness[55,56].

The Eczema Area and Severity Index (EASI)[27], assessed by two board-certified dermatologist, was used for analysis as disease severity. Patient information including disease history, medication history (within 4 weeks for the cross-sectional dataset and 13 months for the longitudinal dataset), laboratory blood test data, and EASI were extracted and filed from electronic medical records along with patient questionnaires.

### Sample collection

For skin RNA-seq, lesional skin biopsy samples (1 mm punch) were obtained from the backs of the participants using Biopsy Punch (Kai Medical) under local anesthesia with Emla creem (lidocaine 2.5% and prilocaine 2.5%, Sato Pharmaceutical) which was administered 1 h before the performance of biopsy. Samples were placed in RNAlater (Life Technologies) overnight at 4 °C and stored at −80 °C until further processing. For immunohistochemistry, skin biopsy samples (1 mm punch) were taken from sites exhibiting similar skin conditions in close proximity (within 5 mm region) to the skin samples for RNA-seq,

immediately snap-frozen and stored at −80 °C until further processing. For PBMC RNA-seq, PBMC were isolated from venous peripheral blood by density gradient purification using Vacutainer CPT tubes (Becton Dickinson) following the manufacturer's instructions, suspended in RNAlater and stored at −80 °C until further processing.

## Immunohistochemistry

We defined the degree of erythema-skewness as erythema/(erythema + papulation) using the EASI partial points, and randomly picked six patients who have erythema-skewness ≥0.6 as erythema-skewed patients and six patients who have erythema-skewness ≤0.4 as papulation-skewed patients for histopathological analysis. Frozen skin samples from the selected AD patients were thawed and immediately embedded in O.C.T. compound (Sakura Finetech), snap-frozen and stored at −80 °C until cryosectioning. Immunostaining was performed using the streptavidin-biotin complex/alkaline phosphatase method as previously described[57] with few modifications. Briefly, 10-μm-thick cryostat-cut tissue sections were fixed for 5 min in ice-cold acetone and rehydrated in phosphate-buffered saline with 0.1% Triton-X followed by incubation with normal goat serum for 1 h. The sections were incubated with the primary antibodies (Table S8) diluted in blocking solution overnight at 4 °C, followed by a biotinylated secondary antibody (either anti-mouse or anti-rabbit according to the primary antibodies, dilution: 1/200) and thereafter with a streptavidin-biotin complex/alkaline phosphatase (Vectastain ABC-AP; Vector). Finally, the sections were developed with alkaline phosphatase substrate (ImmPACT Vector Red; Vector) and counterstained with hematoxylin. The images were captured using a digital image acquisition and analysis system (BX43 microscope, DP27 digital camera, cellSens v3.3 Software; Olympus).

## RNA-seq and data processing

For skin tissue RNA-seq, skin specimens were homogenized with BioMasher (Nippi) in TRIzol Reagent (Thermo), and RNA was isolated with Direct-Zol RNA Kit (ZYMO RESERCH). Library preparation was carried out using NEBNext Ultra RNA Library Prep Kit (New England Biolabs) following the manufacturer's instructions. For PBMC RNA-seq, RNA was isolated using Maxwell 16 LEV simplyRNA Blood Kit and Maxwell 16 Instrument (Promega) and library preparation were carried out with SureSelect Strand-Specific RNA Library Prep Kit (Agilent). The libraries were pooled for skin tissue RNA-seq and PBMC RNA-seq, respectively, and sequenced on HiSeq1500 or HiSeq2500 with bcl2fastq (Illumina) to obtain 15–20 million reads using the 50-bp single-end read configuration. Reads were aligned to the Ensembl GRCh38 human genome assembly using STAR (2.5.2)[58] and feature counts were performed with the R package Rsubread[59]. R version 3.6.2 was used for all the following analysis in R language unless specified otherwise. Genes were filtered by both of the following conditions: (1) expressed in more than 5% of the sample population, (2) maximum reads across the population >8. Samples were filtered with the criteria of total read count > 5 million. Genes coding hemoglobin proteins ("HBA2", "HBB", "HBA1") and ribosomal proteins were removed. The batch effects from each dataset attributable to difference in experimental periods or locations for sequencing were adjusted by ComBat-seq[60] with R package sva. Differential gene expression analysis and vst normalization were conducted using the R package DESeq2[61]. Since there is a chance where skin samples are occupied by considerable volume ratio of pilosebaceous unit in 1 mm punch biopsy, only biased by sampling regions, skin samples were also filtered by gene expression intensity of pilosebaceous unit-related gene set. A cluster that showed extremely strong signature of pilosebaceous unit-related genes in Uniform Manifold Approximation and Projection (UMAP)[62] as analyzed with R package umap, were excluded. GO analysis and GSEA were performed with the R package clusterProfiler[63] and ReactomePA[64].

## Inference in ligand–receptor coupling

Since our datasets consist of bulk-derived samples, which represent mixed signatures of any cell type present in the tissue, we evaluated the degree of ligand–receptor coupling with a binary scoring approach[32] and thereafter cell type specificity for individual active cytokines and receptors were inferred by using publicly available datasets of cell type-specific expression.

Ligand–receptor pairs that are classified into inflammatory response were extracted from the list of cytokine–receptor interactions in the KEGG pathway database (https://www.genome.jp/kegg/)[65]. Possible active cytokine–receptor pairs were defined as concurrent presence of pairs of possible active cytokines and possible active receptors. Considering the biological context for the differential regulation of cytokines and receptors[66] along with previously reported approaches[33], we used the different conditions for the definitions of possible active cytokines and possible active receptors. Possible active cytokines were defined by their expression > 0.5 in the value of vst normalization which accounts for the top 14.2% of the overall population, while possible active receptors were defined by their expression >0 in the value of vst normalization which accounts for the top 48.6% of the overall population.

A total of 210 pairs of inflammatory cytokine and receptor genes were assessed in the skin and PBMC of each of AD patient and healthy control. The active cytokine–receptor pairs were enumerated according to classes defined by the combination of a sender organ that expressed a cytokine gene and a receiver organ that expressed a cognate receptor. Comparison of the number of active connections between cytokines and receptors between AD patients and healthy controls were carried out by a non-parametric Brunner-Munzel rank test[67] with R package lawstat[68], taking into account the nature of the data that showed non-normal and heteroscedastic distribution in two patient groups. P values less than 0.05 were considered significant.

For each of the cytokine and receptor genes, cell types responsible for the cytokine/receptor gene expression were estimated by referring to publicly available datasets (GSE147424; scRNA-seq of skin tissue from AD patients and healthy controls[33], Human Protein Atlas blood cell gene data; RNA-seq of 18 cell types sorted from human peripheral blood[34], for skin and PBMC RNA-seq data, respectively). R package Seurat[69] with R version 4.0.2 was used for scRNA-seq re-analysis. Reference datasets were standardized among cell types and genes that were expressed at a level of z-score >2 were deemed as cell type-specific genes. Note that expression of cytokine/receptor genes were widely shared across multiple cell types in PBMC. Since contribution of granulocytes may be negligible because of their small fraction in PBMC compared to other cell types, we excluded neutrophil, eosinophil and basophil from the cell type annotation in this analysis. Ligand–receptor connection were visualized using the R package circlize[70].

## Module detection and validation

Gene co-expression networks of skin and PBMC transcriptomes were constructed from the vst normalized matrix of variance top 10,000 genes in respective datasets using the R package WGCNA[71]. Modules were generated following the procedures recommended by the publication author, including determination of the algorithm's hyperparameters. Soft-thresholding power ($\beta$) was chosen as the lowest power for which the scale-free topology fit index reached 0.80 with the minimum threshold of 6. As each module is composed of genes highly correlated with each other, the intensity of overall expression of a given module in a patient was represented as the first principal component of expression of all the genes in the module. Hub genes were defined using the signed KME function and transcriptome networks were visualized using the R package igraph[72]. Module characterization was performed based on both cell type specificity and GO. Cell type specificity in its expression was determined by referring to the same

external dataset used in the previous section, i.e. either scRNA-seq (skin) or sorted cell RNA-seq (PBMC). Because number of genes in the PBMC modules was specifically expressed by granulocytes, we included neutrophil, eosinophil, and basophil for the cell type annotation in this analysis. Note that the cell type frequency was not taken into account for the size of the contribution to expression of each gene. GO analysis were performed with the R package clusterProfiler.

## Regression analysis

Elastic net, a regularization and variable selection method that combines the L1 and L2 penalties of the lasso and ridge methods[37], was applied on cross-sectional datasets consisting of both skin and PBMC RNA-seq data along with blood tests (AD patients: $n = 115$, healthy controls: $n = 14$) to determine the strength of the relationship between disease phenotypes and omics features using the R package glmnet[73]. For each phenotype defined with clinical scores, samples were labeled with the degree of specific skin conditions in continuous values, and were split into a training set (70%) and a testing set (30%). Models were built on the training set with optimization of the regularization parameter $\lambda$ which determines how much shrinkage is used to train the model, through ten-fold cross validation. Another hyperparameter of $\alpha$ which determines the ratio of L1 penalty to the combination of L1 and L2 penalties was set to 0.5, intending to exploit both the sparse representation effect in the lasso and the grouping effect in the ridge. Then the model with the optimal parameters was applied to the test set to get the $R^2$ value to evaluate how well the model fit to the observed data.

For longitudinal data analysis, the model was trained on a total cross-sectional dataset excluding 30 AD patients who are enrolled in the longitudinal cohort, and tested on the longitudinal dataset from 30 AD patients. Prediction performance on the test set was evaluated with $R^2$. Closeness of fit in personalized trajectory was evaluated with the Pearson correlation coefficient.

## Longitudinal data analysis

Time series data from blood tests, PBMC transcriptome modules and clinical severity were profiled by patients in date order. By using the Python (version 3.7.4) module Tsfresh[41], seven types of time series features, i.e., mean, minimum, maximum, root mean square (RMS), mean absolute change (MAC), approximate entropy, and complexity-invariant distance (CID) were extracted in individual patients. The values of time series features were standardized among patients. PCA followed by unsupervised k-means clustering was conducted on longitudinal features of PBMC transcriptome modules and blood tests to identify patient clusters based on longitudinal endotypes.

## Reporting summary

Further information on research design is available in the Nature Portfolio Reporting Summary linked to this article.

## Data availability

RNA-seq data generated in this study have been deposited in the National Bioscience Database Center (NBDC) Human Database. Raw data are available at the Japanese Genotype-phenotype Archive (JGA) with accession codes JGAS000628 under controlled access for issue on privacy in informed consent by participants which can be accessed through application for hum0413 at the NBDC. The reference data used in this study are available in the Gene Expression Omnibus database under accession code GSE147424 and Human Protein Atlas database with the title of "RNA HPA immune cell gene data". Source data are provided with this paper.

## Code availability

The source code to reproduce the presented results are available at the online code repository (https://github.com/aico007/AD_heterogeneity_analysis).

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

## Acknowledgements

We would like to sincerely thank all the participants involved in this study. We thank H. Maeo, S. Shibata, R. Sato, M. Tanaka and E. Numazaki for supporting biospecimen sampling. We thank R. Ohashi, A. Hananoe, M. Otsuka, E. Okutsu, Y. Koseki, A. Sugimoto and T. Takemori for supporting maintenance of the storage of human samples and data. We thank R. Edahiro, Y. Tomofuji, S. Koyasu, K. Yamamoto, K. Fujio, T. Endo and T. Ishikawa for helpful advice on analysis. This study was supported by AMED (22ek0410079 and JP19ek0410046, awarded to H. Koseki; JP21ek0410058 and JP18ek0410028, awarded to M.A.), JST (JPMJIH1504, awarded to K.S. and H. Koseki) and Japan Society for the Promotion of Science (JSPS) KAKENHI (18K16072 and 20K17333, awarded to A.S.). The image in Fig. 1 is from TogoTV (© 2016 DBCLS TogoTV, CC-BY-4.0 https://creativecommons.org/licenses/by/4.0/deed.en).

## Author contributions

Study concept and design: H. Koseki, M.A., H. Kawasaki, K.S.; Acquisition of clinical samples: H. Kawasaki, A.F.N., K.Y., K.T., S.T., M.A.; Data collection: K.A., T.M., J.Y., A.K., O.O., H. Kawasaki, A.F.N., A.S.; Analysis and interpretation of data: A.S., Y.O., E.K. J.S., S.N., T.N., Q.W.; Drafting of the paper: A.S., Y.O., H. Koseki. All authors reviewed and approved the final draft of the paper.

## Competing interests

H. Koseki has received research funds (grants paid to his institution) from Maruho and Kao. M.A. has received research support and funds (grants paid to his institution) from Maruho, Ono, Torii, Sato and Taiho. H. Kawasaki has received research funds (grants paid to his institution) from Torii. The rest of the authors declare no competing interests.

## Additional information

[1]RIKEN Center for Integrative Medical Sciences, Yokohama, Japan. [2]Department of Dermatology, Keio University School of Medicine, Tokyo, Japan. [3]Advanced Data Science Project, RIKEN Information R&D and Strategy Headquarters, Tokyo, Japan. [4]Department of Statistical Genetics, Osaka University Graduate School of Medicine, Osaka, Japan. [5]Department of Genome Informatics, Graduate School of Medicine, The University of Tokyo, Tokyo, Japan. [6]Artificial Intelligence Medicine, Graduate School of Medicine, Chiba University, Chiba, Japan. [7]Kazusa DNA Research Institute, Chiba, Japan. [8]Department of Extended Intelligence for Medicine, Keio University School of Medicine, Tokyo, Japan. [9]Cellular and Molecular Medicine, Advanced Research Departments, Graduate School of Medicine, Chiba University, Chiba, Japan. ✉e-mail: yuki-okada@m.u-tokyo.ac.jp; amagai@keio.jp; haruhiko.koseki@riken.jp

