## [Peer Review File · Nature Communications]

Multifaceted analysis of cross-tissue transcriptomes reveals phenotype–endotype associations in atopic dermatitisEditorial Note: Parts of this Peer Review File have been redacted as indicated to maintain the confidentiality of unpublished data. Additionally, patient identifiers have been modified throughout the file in order to maintain patient confidentiality.

REVIEWER COMMENTS

Reviewer #1 (Remarks to the Author):

The paper submitted by Sekita et al. reports on an Omics approach of skin biopsies and PMBC from patients affected by AD with the aim to further mechanistically explore the heterogeneity of the clinical phenotype and thereby to provide some holistic view of this complex disorder. This is an interesting report using cutting-edge technologies which however suffers from some flaws and assumptions that I would like to address briefly.

1. The initial transcriptomic analysis of the 115 AD patients seems to be a retrospective analysis with all the caveats related to such an approach. Before starting such an analysis, the quality of the phenotypic data - including quality of severity scoring (have the investigators been appropriately trained for the use of the EASI score?, the information to the current therapy (and not only the systemic therapy, the local therapy matters as well) at the time point of the sample collection, etc..., - is absolutely crucial for the interpretation of the data. In fact the authors partly recognize this aspect at the end of the results part (page 18, line 377) in stating "that systemic immunosuppressant therapy could partially contribute to instability of disease severity trajectory as well as other personal time series features". I would assume that any kind clinically efficacious immunosuppressive drug such as ciclosporine A (mentioned as a systemic therapy in this study Table S2) must have this impact.

2. The authors are constructing a hypothesis on a phenotypic difference between patients with primarily erythema versus those with a dominant papulation as part of the EASI scoring and try to substantiate this by immunohistochemical analysis. A critical eye on the histopathological slices suggests that there may be 2 other relevant differences between patient AD# [REDACTED] and patient AD# [REDACTED]: first, the presence of vesicles in the first patient and more stratum corneum in the second one strongly suggest tha patient AD# [REDACTED] has been biopsied during an acute flare of his disorder while patient AD# [REDACTED] was more in a chronic phase of the disease. Secondly, the slices of patient AD# [REDACTED] seems to be more tangentially cut and therefore lack the deeper vascular structures. (Another detail with regards to the histopathological pictures: in Figure S3, the bar depicting the magnification of the normal skin seems unrealistic with respect to the overall size of the epidermis and its cells. It must be a different magnification than the other pictures).

3. Thus, if the basic assumption of the 2 distinct clinical phenotypes erythema versus papulation is the product of a misinterpretation of the data, a major part of the working hypothesis and rest of the study are unfortunately questionable although I recognize the tremendous effort done in this study.

4. The authors assume that their study may reflect the AD population but with regards to the mean EASI of their population (EASI 17), the cohort only represents the more moderate and severe patients and not the mild population. This is of significance since most of the dogma around the issue of whether or not AD is a systemic disorder is based on exploration of moderate-to-severe population. In fact, another hypothesis is also valid which is the following: the chronic inflammation of the skin has a systemic impact (instead of AD being primarily a systemic disorder). The interesting analysis of skin to PBMC versus PBMC to skin cytokine and receptors analysis would potentially also speak for this hypothesis.

5. The authors consider the so-called antiallergy drugs as relevant systemic intervention for AD. This is far from being the case as it is well accepted that anti-histamine do not have any therapeutic impact on AD. Most of the international guidelines do not favor their use but they are still prescribed in many countries.

6. It is questionable whether using the longitudinal data of only 30 patients provides the sufficient power for a statistically valid cluster analysis. This should be verified by an expert in biostatistics.

Reviewer #2 (Remarks to the Author):

In this study, authors integrated RNA-seq data from both skin and PBMC along with clinical data from AD patients and matched healthy controls aiming to identify the molecular signature associated with specific clinical presentation/endotypes. Analysis of cross-tissue ligand-receptor coupling suggested increase of skin-PBMC interactions in AD patients compared to healthy

controls. Using combinatorial approach of WGCNA and regression models revealed differential patterns of modular involvement in erythematous and papular skin manifestations in AD and PBMC transcriptomes. They also applied this newly generated regression model approach to a longitudinal dataset.

This study is highly interesting and includes substantial amounts of high-quality data analysis. The novelty of the manuscript is the cross-tissue interactions of modular skin and PBMC transcriptomes with detailed clinical characterisation. Although data analysis seems to be scientifically sound its clinical or biological value is not that clear. Moreover, manuscript is largely descriptive without mechanistic validation and causality analysis.

There are several questions that needs to be addressed properly

1. It seems that there are more analyzed samples than AD patients and controls - e.g. 165 AD patients (271 lesional skin samples and 194 PBMC 111 samples) and 45 healthy controls (56 non-lesional skin samples and 45 PBMC samples). However, it is impossible for the reader to identify how many samples and from which skin location (per patient or per control) etc. are taken and analyzed in different study setups and also why there are more samples than patients and/controls investigated. Clear Table or schematic presentation should be given to clarify how many samples have been analyzed and from which skin location?, duplicates?, repeated samples by time?.
2. I agree that 3 clusters among 30 patients can be identified using EASI time series features, but it is very difficult to see real clinical value of these findings. I would be important to emphasize in the discussion more clearly what is the clinical or biological value of these findings.
3. The study included only 14 healthy controls – why so few?
4. It would be interesting and important to analyze transcriptome profile also from the non-lesional skin of AD patients from the same patients.
5. Why are different methods used in the different figures? Figure2a vs Figure8c (MDS vs PCA)?
6. R2 values are different in Figure 7A compared to corresponding text.
7. In line 210-213, please clarify why a level of cytokine gene > 1 and receptor gene > 0.5 in value of variance stabilizing transformation (vst) are used as the threshold?
8. Line 165 - 177 is repeated, please modify.

Reviewer #3 (Remarks to the Author):

Understanding the heterogeneity in disease phenotypes and endotypes is important for optimal treatment for individual patients. Here, the authors studied the molecular signature associated with clinical phenotypes of atopic dermatitis (AD) by integrating RNA-seq data of both skin and PBMC along with clinical data from 115 AD patients 35 and matched 14 healthy controls. By combining weighted gene co-expression network analysis (WGCNA) and elastic net regression, the authors identified phenotypes-relevant transcriptome modules in skin and PBMC. Based on longitudinal features of blood tests and PBMC transcriptome modules, the authors identified three patient clusters, which were associated with longitudinal features in clinical severity as well as in the medication history.

One of the main conclusion in this study was the important contribution of the identified transcriptome modules to the clinical phenotypes. However, the regression model did not show clear improvement of prediction when adding transcriptome modules as model variables. In many places, method details were not clear. Below are specific comments.

Major points:

1. Fig. 6: what is the difference between linear model and elastic net model? Although the authors claimed better prediction when adding skin and PBMC modules, the improvement is very subtle (0.41 vs. 0.43 for R²). It is not surprising to see such improvement. This result was not sufficient to draw the conclusion that transcriptome modules was closely related with phenotypes. The results still indicate that blood test was the dominant contributor.
2. It is interesting that the authors identified AD-related modules by applying WGCNA to the entire datasets including both AD patients and health control? Can the authors further discriminate the modules related to AD from modules related to health control?
3. Cell types was estimated using scRNA-seq data of AD patients and bulk PBMC RNA-seq. There are plenty of PBMC single cell datasets, Why not use it in Fig 5? In addition, why do not use the same reference cell types of PBMC in fig 4a and fig 5d?
4. The time series features used in this study were single values, which cannot well characterize the dynamic changes over time. Why not use metrics (such as Pearson correlation) that can directly capture the relationship between patients based on the time series data? I am not saying that Pearson correlation is the best metric, but the authors should explore other metrics.

Minor points:

1. What is the meaning of endotypes in this study? In line 407, the authors mentioned that it was defined by the transcriptome modules.
2. Fig 6b: Were the predictors such as Lymphocyte and Eosinophil included in the elastic net model? What are the input values of these variables? What are the meanings of the links in Fig. 6b?
3. Fig. 5b: PC1 value was computed for each module per patient. How were the values related to cell types as shown in Fig. 5b? It was interesting that sModule2 with high PC1 values in keratinocytes was enriched in neuron system. What did this mean in biology?
4. It is not clear how did the PCA was performed in Fig. S4? What is the input of the PCA analysis?
5. Fig. 7: It is confusing to show the axis on the most left. I suggest the authors to show the x/y-axis labels along with the dot plots and only leave the figure legend on the most left panel. The authors mentioned "R²: 0.10 vs. 0.26" in line 320, but these two numbers are not the same as in Fig. 7a. I am very surprised that the calculated R² on the test samples in basic information + blood test was negative when examining the scatter plot in Fig. 7a. Why do not consider skin modules?
6. Fig. 8: Why did the authors only consider the "Mean" and "MAC" when assessing the association of patient clusters with the variables?
7. What is the alpha in line 592?

Response to the reviewer's comments:

Reviewer #1 (Remarks to the Author):

The paper submitted by Sekita et al. reports on an Omics approach of skin biopsies and PMBC from patients affected by AD with the aim to further mechanistically explore the heterogeneity of the clinical phenotype and thereby to provide some holistic view of this complex disorder.

This is an interesting report using cutting-edge technologies which however suffers from some flaws and assumptions that I would like to address briefly.

Response:

We sincerely thank the reviewer for taking time to consider our manuscript and kindly acknowledge the novelty of our study. We appreciate many insightful comments to further improve the manuscript.

1. The initial transcriptomic analysis of the 115 AD patients seems to be a retrospective analysis with all the caveats related to such an approach. Before starting such an analysis, the quality of the phenotypic data - including quality of severity scoring (have the investigators been appropriately trained for the use of the EASI score?, the information to the current therapy (and not only the systemic therapy, the local therapy matters as well) at the time point of the sample collection, etc..., - is absolutely crucial for the interpretation of the data. In fact the authors partly recognize this aspect at the end of the results part (page 18, line 377) in stating "that systemic immunosuppressant therapy could partially contribute to instability of disease severity trajectory as well as other personal time series features". I would assume that any kind clinically efficacious immunosuppressive drug such as ciclosporine A (mentioned as a systemic therapy in this study Table S2) must have this impact.

Response:

Thank you for the insightful comments. We agree with the reviewer's opinion that the quality of the phenotypic data which was used as the dependent variable in the model is crucial for the interpretation of the data.

As to the reviewer's primary concern on the quality of severity scoring, our EASI data was generated by two board-certified dermatologist who had been trained for EASI scoring using the learning video provided by Harmonising Outcome Measures for Eczema (HOME). The dermatologists had regular opportunities to discuss the evaluation criteria with each other before evaluating patients. Therefore, we consider that our phenotypic data meet the standard of dermatology and is qualified enough for the analysis.

For the second point on the current therapy, we admit that both systemic and local therapy could potentially affect the condition of the patient samples. Since this study is observational study aiming to capture the real situation in individual patients, we have recorded all the therapy administered to the patients including topical steroids.

All the AD patients in the studies were basically instructed to use topical steroids on the site where they have dermatitis, with an exception that 5 patients out of 115 patients (4.3%) in the cross-sectional cohort were not prescribed any topical steroids during the study period

for the private reasons. On the other hand, oral steroids and immunosuppressant treatment was in principle, administered only to the deemed recalcitrant patients who don't get sufficient improvements by topical steroids. There were 2 patients (1.7%) and 4 patients (3.4%) in the cross-sectional cohort who were prescribed oral steroids and immunosuppressant, respectively, during the study period. But the number of such patients were small.

As the reviewer pointed out, the influence of these treatment could potentially be a confounding factor in the regression model. However, incorporating such individuals' detailed clinical information into the regression models should drastically increase the number of the categorical predictor variables, which is not feasible for the current sample sizes. Since adjustment by including the covariates is not feasible, we alternatively conducted the subanalysis on the patient subsets that excludes both patients who were not under topical steroids, and who were under oral steroids or immunosuppressant treatment, so that we can rule out the potential influence of treatment difference of the patients.

We found that the most part of the top contributing factors in the original analysis (total 129 subjects) were selected again as contributing factors in the subanalysis (**Table S7**). This suggested that the 11 patients who were in treatment status different from that of the majority of the patients didn't have much influence in our regression models. These results may be partially attributable to the small sample number of the case (patient who are under specific treatment). Larger sample size, for example, ten times to the current dataset, in both the number of the total subjects and the case subjects may highlight the effect of immunosuppressant treatment in our models, which would be our future task.

Modification:

To reflect the reviewer's comments, we modified the description as follows:

"The AD patients in this observational study were basically under treatment with topical steroids and emollients as directed by dermatologists, except for the 5 patients who refrained from using topical steroids for some reason. Their history of systemic treatment was categorized as follows: intermittent use of oral steroids, intermittent use of immunosuppressant, antiallergic agents with continuous use (a total of more than 120 days/year), antiallergic agents with occasional use (a total of fewer than 120 days/year), and no use of these agents. Drugs used for systemic treatment in the overall AD population in this study are listed in the **Table S2**." (**Results**, line 133)

"Erythema was characterized by a bolstered signature of immediate early genes (NR4A1, FOSL1, FOSB, ATF3, NR4A2) and immune system (CD163, C1QB, C1QC, THY1, MS4A7) that are inferentially expressed mainly in keratinocytes and myeloid cells, respectively, in skin tissue, along with Treg specific genes (CCR4, CNTNAP1, DUSP4, LMNA, PI16) in PBMC. In contrast, papulation was characterized by decreased B cell signature (FCRL1, MS4A1, PAX5, CD22, LINC00926) and increased naïve CD4 signature (NELL2, LRRN3, OBSCN, CCR7, GRASP1) in PBMC along with enhanced signature of interferon signaling (MMP12, CCL18, IFI27, TYMP, COL6A6) and extracellular matrix (PI15, GREM1, COL4A1, TNFAIP6, NNMT), suggestive of altered activity in VEC and fibroblast in skin tissue. We confirmed by subanalysis that these results were not biased by the potential influence of the treatment difference among patients (**Table S7, Supplementary note**)." (**Results**, line 308)

"To examine the influence of treatment difference on the results of regression analysis, we conducted subanalysis on the patient subset excluding those who are under the specific treatments. Among the cross-sectional cohort patients, 110 patients (96%) were under

treatment with topical steroids, while 2 patients (1.7%) and 4 patients (3.4%) were under oral steroids and immunosuppressant, respectively, during the study period. Accordingly, the majority of the patients (104 patients, 90.4%) were under treatment only with topical steroids and free from oral steroids or oral immunosuppressant. Therefore, we applied regression analysis on this patient subset along with healthy control (total 118 subjects: 104 AD patients and 14 healthy controls). We found that the large part of the top contributing factors in the original analysis (total 129 subjects) were selected again as contributing factors in the subanalysis (**Table S7**), suggesting that our results of phenotype – endotype association were not biased by the potential influence of the treatment difference among patients.”
(**Supplementary notes**)

Objective variables	Adjusted R^2	Predictors	Coefficient	P-value	Tissue	PC1 top 5 genes	Cell type specificity
EASI (total)	Train 0.63 Test 0.51	Lympho	-0.34	0.0038	Blood	-	Lymphocyte VEC Keratinocyte Eosinophil
		sModu14	0.35	0.024	Skin	MMP12, CCL18, IFI27, TYMP, COL6A6	
		sModu10	0.29	0.071	Skin	S100A8, S100A9, KRT6C, SERPINB4, S100A7	
		Eosino	0.19	0.081	Blood	-	
EASI (erythema)	Train 0.67 Test 0.56	sModu08	0.17	0.027	Skin	NR4A1, FOSL1, FOSB, ATF3, NR4A2	Keratinocyte - Treg
		TARC_log2	0.23	0.041	Blood	-	
		pModu11	0.13	0.069	Blood	CCR4, CNTNAP1, DUSP4, LMNA, PI16	
EASI (papulation)	Train 0.54 Test 0.30	Lympho	-0.35	0.0011	Blood	-	Lymphocyte VEC Eosinophil B cell Naïve CD4 Basophil - - - T cell/myeloid VEC
		sModu14	0.46	0.0019	Skin	MMP12, CCL18, IFI27, TYMP, COL6A6	
		Eosino	0.22	0.0066	Blood	-	
		pModu06	-0.22	0.029	Blood	FCRL1, MS4A1, PAX5, CD22, LINC00926	
		pModu01	0.22	0.040	Blood	NELL2, LRRN3, OBSCN, CCR7, GRASP1	
		pModu03	-0.16	0.052	Blood	PPBP, TUBB1, ITGB3, SDPR, SPARC	
		ALT	0.15	0.055	Blood	-	
		BUN	-0.15	0.056	Blood	-	
		CRP	0.16	0.062	Blood	-	
		sModu05	-0.23	0.070	Skin	LYZ, CCL19, IL7R, RGS1, CCL22	
		sModu16	0.22	0.075	Skin	PI15, GREM1, COL4A1, TNFAIP6, NNMT	

Table S7 Prediction variables extracted by regression analysis applied on the subset of the patients who are under treatment only with topical steroids and free from internal medicine. N: AD = 104, healthy control = 14.

2. The authors are constructing a hypothesis on a phenotypic difference between patients with primarily erythema versus those with a dominant papulation as part of the EASI scoring and try to substantiate this by immunohistochemical analysis. A critical eye on the histopathological slices suggests that there may be 2 another relevant differences between patient AD#1 and patient AD#2: first, the presence of vesicles in the first patient and more stratum corneum in the second one strongly suggest that patient AD#1 has been biopsied during an acute flare of his disorder while patient AD#2 was more in a chronic phase of the disease. Secondly, the slices of patient AD#1 seems to be more tangentially cut and therefore lack the deeper vascular structures. (Another detail with regards to the histopathological pictures: in Figure S3, the bar depicting the magnification of the normal skin seems unrealistic with respect to the overall size of the epidermis and its cells. It must be a different magnification than the other pictures).

Response:

We will respond this comment along with the reviewer's next comment.

3. Thus, if the basic assumption of the 2 distinct clinical phenotypes erythema versus papulation is the product of a misinterpretation of the data, a major part of the working hypothesis and rest of the study are unfortunately questionable although I recognize the tremendous effort done in this study.

Response:

Thank you for the insightful comments. As the reviewer pointed out, we investigated the phenotype-endotype associations based on the hypothesis that phenotypic difference of erythema and papulation can be assessed by using the EASI score and substantiated by histological analysis. We also assumed that the phenotypic difference between erythema and papulation is a distinctive aspect of AD phenotypes from those between acute phase and chronic phase; the feature of erythema/papulation-skewness is rather constitutional within individual and is largely persistent over the disease course. Regarding the disease chronicity, please note that defining chronicity or disease stage in individuals is a grand challenge in various chronic diseases including AD.

Although there currently exist no gold standards regarding the classification of acute phase and chronic phase for AD, findings relevant to disease chronicity in AD have been reported by two groups, which should support our hypothesis that the constitutional features are persistent over the course of acute and chronic phase (Gittler et al., 2012; Tsoi et al., 2020). They examined intrapersonal changes of immune axes from acute lesions to chronic lesions by skin transcriptome analysis, with the definition of acute lesion as (1) new lesions of less than 72 hours' duration (2) lack of skin lichenification and (3) lack of regenerative hyperplasia. Tsoi et al. concluded that the transition from acute to chronic inflammation in AD are quantitative rather than qualitative, proposing a model of progressive activation of immune axes including TH2 and TH22.

Following this model, even if the sample population include skin specimens taken both in acute phase and in chronic phase, intraindividual characteristics are supposed to be qualitatively maintained over the course of inflammation, which warrant our cross-sectional analysis on phenotype-endotype association.

To reinforce our claim on phenotypic differences between erythema and papulation in AD patients, we added the histological data of skin specimens from five erythema-skewed and five papulation-skewed patients, respectively (**Fig. S5**). Additional histological analysis confirmed that both erythema-skewed and papulation-skewed skin were characterized by hyperplasia and immune cell infiltration. The pattern of immune cell infiltration as described in our manuscript were largely reproduced in the additional data; erythema-skewed skin showed diffuse infiltration of immune cells in papillary dermis accompanied by epidermal lymphocytic infiltration (observed in all the samples), while papulation-skewed skin tended to show nodular infiltration of immune cells in dermis.

Based on total 12 skin histological data (6 erythema-skewed and 6 papulation-skewed), we would like to answer to the reviewer's comments on histological observations one by one.

(1) the presence of vesicles in the first patient suggests that patient AD#1 has been biopsied during an acute flare of his disorder.

The circular structure observed around epidermis in the patient AD#1, that looks like a sphere caused by occlusion is not a vesicle. Because of irregular formation of epidermis in the skin of the AD patients especially in their sub-acute-chronic phase, as opposed to regular elongation of rete ridges (psoriasiform) in their highly chronic phase, two dimensional histological images could by chance result in seemingly closed structure within the epidermis depending on the cutting plane. In this case, serial section images of the skin specimen would be helpful for capturing the three-dimensional structure of the skin. We thus added the serial section images of skin specimen of AD#1 in **Fig. S7a**. The vesicle-like structures were tracked with arrows, turning out to be a part of dermal papillae. The similar circular structures were observed in the histological images of AD#13, AD#16, AD#17 AD#18 and AD#20 in **Fig. S5**, all of which turned out to be a part of dermal papillae likewise.

Fig. S7 a. Serial section of skin specimen from a representative erythema-skewed AD patients (AD#1). Different kinds of immunohistochemistry were performed on individual sections.

(2) more stratum corneum in AD#2 suggest that the patient was more in a chronic phase of the disease

Not only AD#2 but all of the five papulation-skewed skin showed hyperkeratosis (accompanied by parakeratosis in AD#16 and acantholysis in AD#17 and AD#20). It is more likely that hyperkeratosis is the fundamental feature of papulation rather than the reflection of chronicity. This feature has been associated also with prurigo-type of AD, a variant subtype of AD which is characterized by nodular skin inflammation, (Folster-Holst et al., 2021), as reasonable enough given the anatomical resemblance between papules and nodules; both papules and nodules are defined as solid or cystic raised spot on the skin, with the difference between them being the size of the spot (MedlinePlus, <https://medlineplus.gov>). Therefore, we described hyperkeratosis as one of the features of papular skin in the manuscript.

(3) the slices of patient AD#1 seems to be more tangentially cut and therefore lack the deeper vascular structures

We confirmed that the lack of the deeper vascular structures in the patient AD#1 was not because of tangential cut but a real nature of the skin specimen. We tried to obtain the sections within 50% inside of the diameter of the semi-cylindrical specimens by dropping first 150-200 μm tangential sections from the surface of specimens, given that diameter of the 1 mm punch biopsy specimens in frozen block become approximately 600 to 800 μm (because of the thickness of a biopsy instrument as well as inevitable shrinkage and stretching of biospecimens during storage and processing). To clarify cutting planes in AD#1 and AD#2 in **Fig. 2b** where we parallelly showed equivalent sections in the two patients, we added images of the serial sections of AD#2 in **Fig. S8b** along with AD#1 as already mentioned above, so that vascularization (CD31 immunostaining is especially helpful) and perivascular cell infiltration are generally captured three-dimensionally.

Fig. S7 b. Serial section of skin specimen from a representative population-skewed AD patients (AD#2). Different kinds of immunohistochemistry were performed on individual sections.

(4) in Figure S3, the bar depicting the magnification of the normal skin seems unrealistic with respect to the overall size of the epidermis and its cells. It must be a different magnification than the other pictures.

We confirmed that the scale bars in **Fig. S3** reflects the real scale, and therefore, the contrast between AD and the healthy control in epidermal thickness is real. To facilitate recognizing their actual sizes, we added the low-magnification images of all of AD#1, AD#2 and HC#051 in **Figure S3**. Parts of sweat glands appeared in three images, either in its semi-transverse plane (AD#1 and HC#2) or semi-vertical plane (AD#2) of the ducts would be helpful for relative comparisons among three specimens.

Fig. S6 Immunohistochemical analysis revealed shared and differential characteristics in the skin tissue of erythema and papulation-skewed AD patients.

Modification:

To reflect the reviewer's comments, we modified the manuscript as follows;

“In order to pathologically characterize both erythema and papulation skin in AD, we conducted immunohistochemistry of lesional skin from the six erythema-skewed and the six papulation-skewed patients (**Fig. S5-7, Fig. 2b-c**). **Fig. 2b-c** shows clinical and histological images of the representative patients who have a score composition that is highly skewed to either erythema or induration/papulation (partial score for the left patient; erythema = 9.6, papulation = 4.8, the right patient; erythema = 4.3, papulation = 8.6). Histological analysis revealed shared and differential characteristics in the skin tissue between the erythema- and papulation-skewed AD patients.” (**Results**, line 165)

“However, the patterns for immune cell infiltration appeared to be different between the two skin samples; the skin sample from the erythema-skewed patient were characterized by

diffuse infiltration of immune cells in dermis, accompanied by epidermal lymphocytic infiltration (**Fig. 2c** right panel). On the other hand, the skin sample from the population-skewed patient was characterized by nodular infiltration of immune cells in dermis suggestive of geometrical heterogeneity over the lesion, as well as prominent hyperkeratosis. Those observations were largely reproduced in other five erythema-skewed patients and five population-skewed patients, respectively (**Fig. S6**).” (**Results**, line 181)

“We defined the degree of erythema-skewness as erythema/(erythema + papulation) using the EASI partial points, and randomly picked six patients who have erythema-skewness ≥ 0.6 as erythema-skewed patients and six patients who have erythema-skewness ≤ 0.4 as population-skewed patients. Frozen skin samples from the selected AD patients were thawed and immediately embedded in O.C.T. compound (Sakura Finetech), snap-frozen and stored at -80°C until cryosectioning.” (**Methods**, line 532)

a

b
(Erythema-skewed AD)

AD#11

AD#12

AD#13

AD#14

AD#15

c

(Population-skewed AD)

AD#16

AD#17

AD#18

AD#19

AD#20

Fig. S5 Immunohistochemistry of skin tissue from AD patients who have either erythema- or population-skewed skin manifestations.

a. We defined the degree of erythema-skewness as erythema/(erythema + population) using EASI partial points, and randomly picked 5 patients who have erythema-skewness ≥ 0.6 as erythema-skewed patients and 5 patients who have erythema-skewness ≤ 0.4 as population-skewed patients. **b,c.** Immunohistochemistry of skin tissue stained for CD4 (target protein was stained in red) in 5 erythema-skewed patients (**b**) and 5 population-skewed patients (**c**). Bars: 500 μm (the right column), 100 μm (the middle and left columns).

4. The authors assume that their study may reflect the AD population but with regards to the mean EASI of their population (EASI 17), the cohort only represents the more moderate and severe patients and not the mild population. This is of significance since most of the dogma around the issue of whether or not AD is a systemic disorder is based on exploration of moderate-to-severe population. In fact, another hypothesis is also valid which is the following: the chronic inflammation of the skin has a systemic impact (instead of AD being primarily a systemic disorder). The interesting analysis of skin to PBMC versus PBMC to skin cytokine and receptors analysis would potentially also speak for this hypothesis.

Response:

Thank you for the insightful comments. The frequency of the patients in each class of EASI-based strata proposed by Chopra et al. (mild: 0.1 – 5.9, moderate: 6.0 – 22.9, severe: 23-72) (Chopra et al., 2017) in our cross-sectional dataset was as follows: mild = 19 (16.5%), moderate = 63 (54.8%) and severe = 33 (28.7%) (**Fig. S3**), resulting in being substantially focused on moderate and severe patients similar to most of the previous cohorts on AD, as the reviewer concerned.

Fig. S3 Histogram of frequency of AD patients at different score of EASI. Severity strata was defined using EASI in individual patients; mild: 0.1 – 5.9, moderate: 6.0 – 22.9, severe: 23-72.

Regarding the discussion on whether AD is a systemic disorder or not, we consider the reviewer’s hypothesis as possible; the chronic inflammation of the skin may have a systemic

impact. We used the word “a systemic disease” in our manuscript to characterize AD with the implication that systemic diseases include the diseases that show secondary alterations in multiple organs or in bloodstream caused by some primary and localized inflammations.

While most of the previous studies have focused on moderate-to-severe AD, as the reviewer has pointed out, He et al. reported that mild AD patients showed inflammation only in the localized skin lesions and lack systemic inflammation differently from moderate or severe AD (He et al., 2021). This poses the possibility that not mild AD but only moderate-to-severe AD is a systemic disease. Another possibility is that AD is basically a systemic disease, but the systemic alterations become detectable if the patients’ severity go over a certain threshold from mild. Our stratified analysis on skin – PBMC interaction showed progressive augmentation of the number of links in severe AD compared to moderate and mild AD. This suggests that systemic inflammation is more evidently involved in severe AD, although there was no statistical difference among three groups (Kruskal-Wallis test, $p = 0.07$).

Fig. S8 Comparison of number of active connections between skin and PBMC among three AD severity strata.

The number of active connections were assessed based on cytokine – receptor coupling. Severity strata was defined using EASI in individual patients; mild: 0.1 – 5.9, moderate: 6.0 – 22.9, severe: 23-72. Multiple comparison tests were carried out with Kruskal-Wallis test.

One of the reasons for the populational imbalance in severity classes could be the fact that the patients in our cohort were enrolled in the single university hospital. Given that AD is not a emergent life-threatening disease but a longstanding QOL-affecting disease with patients’ QOL being dependent on the disease severity which in turn likely to influence how seriously the patients take their medication, the AD patient who choose to visit the university hospital may have some tendency in terms of disease severity and self-awareness. We had included this population bias in the description “limited sample size and population diversity” (**Discussion**, line 469), but it was not clear enough. Therefore, we added the description on this point in **Discussion**.

Please note that beyond these potential confounding factors, our data is highly valuable resource as the first cross-tissue and longitudinal transcriptome of non-European AD.

Modification:

To reflect the reviewer's comments, we modified the description on study limitation as follows:

"Patients were further filtered by gene expression intensity of pilosebaceous unit-related genes in skin samples (**Fig. S2**), resulting in 115 AD patients and 14 healthy controls as eligible samples (one sample per patient) for regression analysis using all of skin, PBMC and blood tests. Frequency distribution of the AD patients by disease severity is shown in **Fig. S3**." (**Results**, line 116)

"Other limitations in our study includes limited sample size and population diversity, as is always the challenge in studies on complex human diseases. Above all, the AD patients in our cohort were enrolled in the single university hospital and can be potentially characterized by specific spectrum in disease severity. Studies with extended sample size and diversity may illuminate more profound heterogeneity in AD." (**Discussion**, line 468)

"Among these, the number of connections from skin to skin and the number of connections from skin to PBMC were significantly increased in AD patients compared to healthy controls (mean = 24.6 vs 10.9 and 17.3 vs 10.6; $p = 8.3E-5$ and $p = 2.8E-3$, respectively), while the number of connections from PBMC to either of skin or PBMC was not significantly different between AD patients and healthy controls (**Fig. 4b**). Stratified analysis on skin – PBMC interaction revealed progressive augmentation of the number of links in severe AD compared to moderate and mild AD, suggesting that systemic inflammation is more evidently involved in severe AD, although there was no statistical difference among three groups (**Fig. S8**). There were moderate correlations between the total number of cytokine – receptor connections and either EASI ($r = 0.32$; $p = 2.5E-4$) or serum TARC ($r = 0.35$; $p = 6.6E-5$, **Fig. 4c**)." (**Results**, line 218)

5. The authors consider the so-called anti-allergy drugs as relevant systemic intervention for AD. This is far from being the case as it is well accepted that anti-histamine do not have any therapeutic impact on AD. Most of the international guidelines do not favor their use but they are still prescribed in many countries.

Response:

Thank you for the insightful comment. As the review pointed out, we included anti-allergy drugs (mostly antihistamine) in systemic medication of the AD patients. We currently realize the generally accepted notion that antihistamine does not have much therapeutic impact on AD. However, around the period we conducted sampling which is from 2016 to 2020, the antihistamines were considered to have a positive effect on relieving itching, and therefore were recommended as an adjuvant therapy to anti-inflammatory topical therapy for AD by the Japanese guidelines for atopic dermatitis (Kato et al., 2020). The recommendation was made based on the evidence of randomized control trials on an adjuvant use of antihistamine in AD patients, some of which was carried out in Japan (Kawashima et al., 2003). Therefore, the dermatologists in our study prescribed antihistamines to the patients according to the complaint of pain by the patients.

This policy on antihistamine usage in the Japanese guideline changed in 2021: the level of recommendation on an adjuvant use of antihistamine was reduced from previous "Recommendation grade 1: recommended" to "Recommendation grade 2: suggested" (Saeki

et al., 2022) because of uncertainty of its efficacy on relief of itchiness (Matterne et al., 2019). Following the new guideline, it is expected that frequency of antihistamine prescription for AD patients in Japan will decrease in the coming years, but we consider that this change in frequency of antihistamine use may not bias the results of our study.

Modification:

For the better clarification, we modified the manuscript as follows:

“All the patients included in two analyses were treated according to the Japanese guideline for atopic dermatitis [54], such as emollients, topical corticosteroids and/or tacrolimus, oral antihistamines and immunosuppressants [29]. Note that the use of antihistamines was recommended as an adjuvant therapy to anti-inflammatory topical therapy to reduce itchiness in the treatment policy proposed by the Japanese guideline at the moment (i.e. 2016-2020) [54]. This policy was later modified to lower the grading of recommendation for the use of antihistamines in the revised guideline in 2021 in response to the increased recognition of uncertainty of its efficacy on relief of itchiness [55, 56].” (**Methods**, line 503)

6. It is questionable whether using the longitudinal data of only 30 patients provides the sufficient power for a statistically valid cluster analysis. This should be verified by an expert in biostatistics.

Response:

Thank you for the practical comment. We consider that analysis on time series data is quite essential given the heterogeneity in disease trajectories in AD population. On the other hand, it is generally challenging to estimate statistical power in such analysis. Please note that it is quite hard even for experts in biostatistics to determine the correct number of sample size in this kind of study, since there is no precedent longitudinal study that involves repeated sampling of biospecimens from AD patients throughout a year. For time series analysis, the power is not simply determined by the sample size but determined by a combined effect of patient number, number of time points, interval, total period and so on. Currently, there may not exist recommendation for the priors of such factors.

In our longitudinal analysis, we aimed to capture intra-patient variation of molecular signatures in the AD patients that has never been examined before, expecting to set a milestone for future studies. We considered sample size of 30 would serve for this purpose. Actually, 30 was one of the the maximum numbers that one can practically afford in the year-wide clinical research. Since our longitudinal analysis was mostly not aiming discovery of new biological mechanism, the typical sample sizes that may be required in the discovery phase such as hundreds or thousands, may not be necessary here.

Further, there is no generally accepted rules for minimally required sample sizes for cluster analysis (Siddiqui, 2013). Even though several indices such as Dunn index, are used to evaluate the separation or compactness of clusters, no guidance on validity of clustering results with given value of index are available.

In our data of 30 patients, Dunn index which is defined as the ratio of the smallest inter-cluster distance to the largest intra-cluster distance (Dunn, 1973), for k-means clustering was 0.66, while that for classification with treatment status was 0.53, suggesting that our clustering results achieved compactness comparable to clinical classification. Adjusted rand

index (ARI), which evaluate similarity of two clustering outcomes, between k-means clustering and classification with treatment status was 0.29, while ARI between k-means clustering and random allocation was 0.033 (an average of 1000 different seeds), and ARI between the label of treatment status and random allocation was 0.037. This suggests that the result of k-means clustering was closer to clinical classification than random allocation. Collectively, although it might not be the “best” clustering result with high marks of separation and compactness, our clustering result have consistency in both views of bioinformatics and clinical significance. We are glad if the reviewer kindly accepts our standpoint.

References

- Chopra, R., Vakharia, P.P., Sacotte, R., Patel, N., Immaneni, S., White, T., Kantor, R., Hsu, D.Y., and Silverberg, J.I. (2017). Severity strata for Eczema Area and Severity Index (EASI), modified EASI, Scoring Atopic Dermatitis (SCORAD), objective SCORAD, Atopic Dermatitis Severity Index and body surface area in adolescents and adults with atopic dermatitis. *Br J Dermatol* 177, 1316-1321. 10.1111/bjd.15641.
- Dunn, J.C. (1973). A Fuzzy Relative of the ISODATA Process and Its Use in Detecting Compact Well-Separated Clusters. *Journal of Cybernetics* 3, 32-57. 10.1080/01969727308546046.
- Folster-Holst, R., Reimer, R., Neumann, C., Proksch, E., Rodriguez, E., Weidinger, S., Goldust, M., Hanisch, E., Dahnhardt-Pfeiffer, S., and Freitag-Wolf, S. (2021). Comparison of Epidermal Barrier Integrity in Adults with Classic Atopic Dermatitis, Atopic Prurigo and Non-Atopic Prurigo Nodularis. *Biology (Basel)* 10. 10.3390/biology10101008.
- Gittler, J.K., Shemer, A., Suarez-Farinas, M., Fuentes-Duculan, J., Gulewicz, K.J., Wang, C.Q., Mitsui, H., Cardinale, I., de Guzman Strong, C., Krueger, J.G., and Guttman-Yassky, E. (2012). Progressive activation of T(H)2/T(H)22 cytokines and selective epidermal proteins characterizes acute and chronic atopic dermatitis. *J Allergy Clin Immunol* 130, 1344-1354. 10.1016/j.jaci.2012.07.012.
- He, H., Del Duca, E., Diaz, A., Kim, H.J., Gay-Mimbrera, J., Zhang, N., Wu, J.N., Beaziz, J., Estrada, Y., Krueger, J.G., et al. (2021). Mild atopic dermatitis lacks systemic inflammation and shows reduced nonlesional skin abnormalities. *J Allergy Clin Immunol* 147, 1369-1380. 10.1016/j.jaci.2020.08.041.
- Katoh, N., Ohya, Y., Ikeda, M., Ebihara, T., Katayama, I., Saeki, H., Shimojo, N., Tanaka, A., Nakahara, T., Nagao, M., et al. (2020). Japanese guidelines for atopic dermatitis 2020. *Allergol Int* 69, 356-369. 10.1016/j.alit.2020.02.006.
- Kawashima, M., Tango, T., Noguchi, T., Inagi, M., Nakagawa, H., and Harada, S. (2003). Addition of fexofenadine to a topical corticosteroid reduces the pruritus associated with atopic dermatitis in a 1-week randomized, multicentre, double-blind, placebo-controlled, parallel-group study. *Br J Dermatol* 148, 1212-1221. 10.1046/j.1365-2133.2003.05293.x.
- Matterne, U., Bohmer, M.M., Weisshaar, E., Jupiter, A., Carter, B., and Apfelbacher, C.J. (2019). Oral H1 antihistamines as 'add-on' therapy to topical treatment for eczema. *Cochrane Database Syst Rev* 1, CD012167. 10.1002/14651858.CD012167.pub2.
- Saeki, H., Ohya, Y., Furuta, J., Arakawa, H., Ichiyama, S., Katsunuma, T., Katoh, N., Tanaka, A., Tsunemi, Y., Nakahara, T., et al. (2022). English Version of Clinical Practice Guidelines for the Management of Atopic Dermatitis 2021. *J Dermatol* 49, e315-e375. 10.1111/1346-8138.16527.
- Siddiqui, K.A. (2013). Heuristics for Sample Size Determination in Multivariate Statistical Techniques.
- Tsoi, L.C., Rodriguez, E., Stolzl, D., Wehkamp, U., Sun, J., Gerdes, S., Sarkar, M.K., Hubenthal, M., Zeng, C., Uppala, R., et al. (2020). Progression of acute-to-chronic atopic dermatitis is associated with quantitative rather than qualitative changes in cytokine responses. *J Allergy Clin Immunol* 145, 1406-1415. 10.1016/j.jaci.2019.11.047.

Reviewer #2 (Remarks to the Author):

In this study, authors integrated RNA-seq data from both skin and PBMC along with clinical data from AD patients and matched healthy controls aiming to identify the molecular signature associated with specific clinical presentation/endotypes. Analysis of cross-tissue ligand-receptor coupling suggested increase of skin-PBMC interactions in AD patients compared to healthy controls. Using combinatorial approach of WGCNA and regression models revealed differential patterns of modular involvement in erythematous and papular skin manifestations in AD and PBMC transcriptomes. They also applied this newly generated regression model approach to a longitudinal dataset.

This study is highly interesting and includes substantial amounts of high-quality data analysis. The novelty of the manuscript is the cross-tissue interactions of modular skin and PBMC transcriptomes with detailed clinical characterisation. Although data analysis seems to be scientifically sound, its clinical or biological value is not that clear. Moreover, manuscript is largely descriptive without mechanistic validation and causality analysis.

Response:

We sincerely thank the reviewer for taking time to consider our manuscript and kindly acknowledging the value of our study on phenotype-endotype association in AD. We appreciate many insightful comments to further improve the manuscript.

There are several questions that needs to be addressed properly

1. It seems that there are more analyzed samples than AD patients and controls - e.g. 165 AD patients (271 lesional skin samples and 194 PBMC 111 samples) and 45 healthy controls (56 non-lesional skin samples and 45 PBMC samples). However, it is impossible for the reader to identify how many samples and from which skin location (per patient or per control) etc. are taken and analyzed in different study setups and also why there are more samples than patients and/controls investigated. Clear Table or schematic presentation should be given to clarify how many samples have been analyzed and from which skin location?, duplicates?, repeated samples by time?.

Response:

Thank you for the kind comment and suggestion. According to the reviewer's suggestion, we added the schematic presentation showing the process of filtering samples and patients for each analysis in **Fig. S1**.

Modification:

“Characterization of participants

A schematic presentation of the process of filtering samples and patients for each analysis was shown in **Fig. S1**.” (Results, line 110)

Fig. S1 A schematic presentation showing the process of filtering sample and patient for each analysis.

2. I agree that 3 clusters among 30 patients can be identified using EASI time series features, but it is very difficult to see real clinical value of these findings. It would be important to emphasize in the discussion more clearly what is the clinical or biological value of these findings.

Response:

Thank you for the valuable comment. As the reviewer pointed out, it is important to clearly propose the clinical and biological significance in the longitudinal part. For the clinical significance, we consider that identifying patient clusters is highly meaningful as the first step toward personalized medicine. Identification of three patient clusters in our longitudinal analysis is practically interpretable. For biological significance, gene modules associated with the identified patient clusters would provide pathological insights of the patient clusters at the molecular level.

Modification:

According to the reviewer's suggestion, we added the description on the biological and clinical significance in the Discussion part as follows:

“With an increase of therapeutic options expected in the coming years, (1) understanding heterogeneity in disease phenotypes and endotypes and (2) patient stratification into subgroups based on phenotypes or endotypes, are the two urgent tasks for the development of personalized medicine in AD.” (**Discussion**, line 401)

“Furthermore, we assessed heterogeneity in the personal longitudinal features in PBMC transcriptome modules and blood tests in association with clinical severity. We identified three patient clusters based on the longitudinal blood-derived signatures that were found to be closely linked with disease course and medication history. Our demonstration is the first step of patient stratification in the view of longitudinal features in AD, serving as a significant movement toward the grand challenge of personalized medicine.

There were also biological findings in the longitudinal analysis. Three top contributing factors for patient clustering was pModu07, pModu09, and neutrophil count, all of which was signatures reflecting innate immunity activity. This suggested that the dynamics of innate immunity may be the major force for instability in longitudinal disease course. As to the factors correlated with disease severity in individual patients, in addition to serum TARC, LDH and eosinophil counts, all of which are well-recognized biomarkers in AD [47], newly defined PBMC transcriptome modules including pModule01 (inferred cell specificity: naïve CD4, PC1 top genes: *NELL2*, *LRRN3*, *OBSCN*, *CCR7* and *GRASP1*) and pModu04 (inferred cell specificity: Treg, PC1 top genes: *MKI67*, *RRM2*, *TOP2A*, *ASPM* and *MYBL2*) were identified as contributing factors in a personal disease course.” (**Discussion**, line 434)

3. The study included only 14 healthy controls – why so few?

Response:

Please note that sampling of biospecimens, especially invasive sampling like skin biopsy, from healthy subjects is generally challenging in terms of ELSI. Additionally, acquisition of blood test data is a routine clinical work in hospitals but not outside hospitals. This likely yield missing clinical values in the healthy controls, making individual data incomplete and be excluded from regression analysis. In fact, we collected biospecimens from over 40 healthy subjects, but we excluded more than half of them because of missing value in blood tests. (Please refer to the **Fig. S1** of the schematic presentation of sample filtering as added in response to the reviewer's question 1.)

Nonetheless, we would like to emphasize that our study was intended to highlight the differential features or heterogeneity within the AD patients, rather than overall AD features which may be elucidated by the case-control comparative analysis. In the analysis, we handled the severity of AD as a continuous scale while setting the value for the healthy control group as 0 in regression models. This can effectively enhance the statistical power than those of the binary scale models for the case-control comparisons in the previous studies.

4. It would be interesting and important to analyze transcriptome profile also from the non-lesional skin of AD patients from the same patients.

Response:

Thank you for the valuable comment. We agree with the reviewer's comment on the importance of analyzing non-lesional skin of AD patients, since non-lesional skin would inform about congenital epidermal barrier function or immune regulation in pre-disease states. However, we decided not to include non-lesional skin of AD in our analysis for the following two reasons.

Firstly, some patients, especially those characterized by erythema-skewed skin manifestations with diffusely distributed dermatitis, do not have non-lesional skin area on their back (please refer to the Figure below for an example), thereby lacking the non-lesional skin samples. In addition to sample filtering by pilosebaceous unit inclusion as mentioned in the **Methods** in non-lesional skin (if obtained), dropping the patients who lack the non-lesional skin would result in further reduction in sample size to the 83 AD patients from the original sample size of the 115 AD patients. Moreover, the sample population may be biased by yet-to-be-defined confounding factors, likely to compromise our aim to illuminate patient heterogeneity. Therefore, we considered inclusion of non-lesional skin in the analysis lead to a huge loss in sample size thereby disadvantageous to the analysis.

[REDACTED]

Secondly, including non-lesional skin data in the regression model would drastically diminish the size of events per variable (EPV: the ratio of sample number to variable number) to 1.33 from original 2.35 in the model without non-lesional skin. Although elastic net was initially touted as the technique applicable to high-dimensional data with small sample size, there seemed to be not much precedent for regularized regression models applied on data with EPV less than 2 (Ambler et al., 2012). Furthermore, recent studies have raised concern on uncertain reliability for prediction performance of the regularized model when effective sample size are low (Riley et al., 2020). Collectively, we considered that inclusion of non-lesional skin samples in the analysis would lead to lower reliability in the results.

Just for reviewer's reference, we nonetheless tested elastic net regression on data including non-lesional skin sample (**Table exclusively for the reviewer**). Among skin modules in non-lesional skin, sModu20, sModu09 and sModu05 were selected for significant variables. These factors might be contributing to the phenotypes, but this should be assessed on data with larger sample size in the future. For now, we refrain from reporting the temporal results in this paper.

[REDACTED]

Modification:

To reflect the reviewer's comments, we modified the description as follows:

“Other limitations in our study includes limited sample size and population diversity, as is always the challenge in studies on complex human diseases. Above all, the AD patients in our cohort were enrolled in the single university hospital and can be potentially characterized by specific spectrum in disease severity. Studies with extended sample size and diversity may illuminate more profound heterogeneity in AD. Including non-lesional skin in the analysis would also serve this purpose since non-lesional skin could be a representation for congenital epidermal barrier function or immune regulation in pre-disease states.” (**Discussion**, line 468)

5. Why are different methods used in the different figures? Figure2a vs Figure8c (MDS vs PCA)?

Response:

Thank you for the insightful comment. As the review pointed out, **Fig. 2a** is a MDS plot on individual components of EASI across AD patients, and **Fig. 8c** is a PCA plot on time series features of both blood tests and PBMC transcriptome modules in 30 AD patients.

MDS and PCA are both unsupervised learning methods widely used for dimension reduction. While MDS is a visual representation of distances between sets of objects (Mead, 1992), PCA provides the data distribution on the projection of directions that explain the variability in the data. The input of MDS should be the pairwise distances between points, while that of PCA should be the original vectors in n -dimensional space. In brief, MDS is suitable to visualize relative distances across the samples, and PCA has an advantage to visualize sample distribution based on the underlying structure.

In **Fig. 2a**, we aimed to clarify relationship among 16 components (4 skin manifestations \times 4 body regions) of EASI score. A set of those components is derived from each patient, and therefore, the components would show association with each other when thousands of sets were analyzed across patients (total 1424 data from 151 patients at different time points were used). We considered MDS is the most suitable method for visualizing the level of similarity among individual components.

In **Fig. 8c**, on the other hand, we aimed to overview the population using all the blood derived factors and identify patient clusters that have differential features. In PCA, PC1 represents the maximum variance direction in the data, and PC2 represents the direction orthogonal to the PC1. We considered that a PC1-PC2 plot is the finest visualization for capturing the patient distribution based on an underlying structure that are differential across patients. Additionally, as Ding et al. have reported that “principal components are the continuous solutions to the discrete cluster membership indicators for K-means clustering” (Ding and He, 2004), the concordance between PCA and K-means clustering is high, warranting our way of visualization where K-means clustering were projected onto the PCA plot.

Modification:

For the better clarification, we modified the manuscript as follows.

“To capture the relationship between individual components of eczema severity, we performed multidimensional scaling (MDS) which is a visual representation of distances between sets of objects [28] on the collection of partial scores across patients (**Fig. 2a**).” (**Results**, line 151)

“Unsupervised k-means clustering on 30 AD patients based on time series features of PBMC transcriptome modules and blood tests, with number of clusters (= k) determined using silhouette criterion, identified three patient clusters (**Fig. 8b**). We applied PCA to this data of time series features to capture the patient distribution in a reduced dimension based on the underlying structure that are differential across patients (**Fig. 8c**), and evaluated the intensity of the top PC1/PC2 contributing factors (**Fig. 8d**).” (**Results**, line 366)

6. R2 values are different in Figure 7A compared to corresponding text.

Response:

Thank you for letting us know our typo. The value in **Fig. 7A** is the correct value and the value in the corresponding text is wrong. We sincerely modified the description in the manuscript.

7. In line 210-213, please clarify why a level of cytokine gene > 1 and receptor gene > 0.5 in value of variance stabilizing transformation (vst) are used as the threshold?

Response:

Thank you for asking this important point. As the reviewer pointed out, we evaluated the degree of cytokine – receptor coupling based on a combined condition of cytokine gene > 0.5 and receptor gene > 0 in their expression in vst value. (There was a typo in the description in the **Results** section. The conditions we used is cytokine gene > 0.5 and receptor gene > 0 , as described in the **Methods** section. Therefore, we updated the description in the **Results** section.)

Efficiency of ligand – receptor interactions which imply molecular binding followed by downstream signaling, may not be determined by simple expression levels of ligand and receptor genes. It would rather be determined by combined effects of various regulation mechanisms at different levels such as proteolytic cleavage (Afonina et al., 2015), receptor internalization and endocytic sorting towards degradation or recycling (Cendrowski et al., 2016). Thus, analyzing efficiency of ligand – receptor interaction in a given tissue is quite a tough challenge and therefore, simple models to infer just the chances of ligand – receptor binding based on gene expression are widely used. Among them, the most frequently used technique is the binary scoring method where active cytokine – receptor pairs are defined as pairs that show gene expression level of both ligands and receptors above certain thresholds (Armingol et al., 2021).

Although using the same value for threshold of both ligands and receptors may be the simplest idea, there is no biological validity for this idea. The facts biologically relevant to this issue would be (1) ligands diffuse in extent of extracellular space while receptors sit on cell surface, suggesting that higher level of gene expression change would be required for ligands than receptors to elicit the same level of upregulation in ligand – receptor interactions, and (2) some receptors will be recycled back to the cell surface after ligand binding and endocytosis (Marchese, 2014), indicating that some receptors are not fully regulated by gene expression but may be more flexibly engaged in ligand – receptor interactions depending on the biological context. Considering those factors, it is more natural to set a higher threshold of gene expression for ligands compared to receptors. Therefore, we used the thresholds of > 0.5 vst for ligands and > 0 vst for receptors.

Similar strategies were used in several studies that include analysis on ligand – receptor interaction, presumably based on the same biological contexts we took into account; He et al. used the thresholds of > 3 TPM for ligands and > 0.4 TPM for receptors (He et al., 2020); Choi et al. used threshold of FPKM > 2 , fold change > 1.5 , and adjusted p value < 0.1 for ligands, and FPKM values > 2 for receptors (Choi et al., 2015).

Therefore, we consider our methods is acceptable in the field of patient-level biology. In the future, however, a more detailed model on ligand – receptor interaction where other confounding factors such as receptor recycling or downstream signaling are quantitatively analyzed, would be preferable.

Modification:

To reflect the reviewer’s comments, we modified the description as follows:

“Ligand-receptor pairs that are classified into inflammatory response were extracted from the list of cytokine - cytokine receptor interactions in the KEGG pathway database (<https://www.genome.jp/kegg/>) [64]. Possible active cytokine - receptor pairs were defined as concurrent presence of pairs of possible active cytokines and possible active receptors. Considering the biological context for the differential regulation of cytokines and receptors [65] along with previously reported approaches [33], we used the different conditions for the definitions of possible active cytokines and possible active receptors. Possible active cytokines were defined by their expression > 0.5 in the value of vst normalization which accounts for the top 14.2% of the overall population, while possible active receptors were defined by their expression > 0 in the value of vst normalization which accounts for the top 48.6% of the overall population.” (Methods, line 581)

8. Line 165 - 177 is repeated, please modify.

Response:

Thank you for letting us know our typo. We corrected the manuscript accordingly.

References

- Afonina, I.S., Muller, C., Martin, S.J., and Beyaert, R. (2015). Proteolytic Processing of Interleukin-1 Family Cytokines: Variations on a Common Theme. *Immunity* 42, 991-1004. 10.1016/j.immuni.2015.06.003.
- Ambler, G., Seaman, S., and Omar, R.Z. (2012). An evaluation of penalised survival methods for developing prognostic models with rare events. *Stat Med* 31, 1150-1161. 10.1002/sim.4371.
- Armingol, E., Officer, A., Harismendy, O., and Lewis, N.E. (2021). Deciphering cell-cell interactions and communication from gene expression. *Nature Reviews Genetics* 22, 71-88. 10.1038/s41576-020-00292-x.
- Cendrowski, J., Maminska, A., and Miaczynska, M. (2016). Endocytic regulation of cytokine receptor signaling. *Cytokine Growth Factor Rev* 32, 63-73. 10.1016/j.cytogfr.2016.07.002.
- Choi, H., Sheng, J., Gao, D., Li, F., Durrans, A., Ryu, S., Lee, S.B., Narula, N., Rafii, S., Elemento, O., et al. (2015). Transcriptome analysis of individual stromal cell populations identifies stroma-tumor crosstalk in mouse lung cancer model. *Cell Rep* 10, 1187-1201. 10.1016/j.celrep.2015.01.040.
- Ding, C., and He, X. (2004). K-means clustering via principal component analysis. Proceedings of the twenty-first international conference on Machine learning. Association for Computing Machinery.
- He, H.L., Suryawanshi, H., Morozov, P., Gay-Mimbrera, J., Del Duca, E., Kim, H.J., Kameyama, N., Estrada, Y., Der, E., Krueger, J.G., et al. (2020). Single-cell transcriptome analysis of human skin identifies novel fibroblast subpopulation and enrichment of immune subsets in atopic dermatitis. *J Allergy Clin Immunol* 145, 1615-1628. 10.1016/j.jaci.2020.01.042.
- Marchese, A. (2014). Endocytic trafficking of chemokine receptors. *Curr Opin Cell Biol* 27, 72-77. 10.1016/j.ceb.2013.11.011.
- Mead, A. (1992). Review of the Development of Multidimensional Scaling Methods. *Journal of the Royal Statistical Society. Series D (The Statistician)* 41, 27-39. 10.2307/2348634.
- Riley, R.D., Ensor, J., Snell, K.I.E., Harrell, F.E., Jr., Martin, G.P., Reitsma, J.B., Moons, K.G.M., Collins, G., and van Smeden, M. (2020). Calculating the sample size required for developing a clinical prediction model. *BMJ* 368, m441. 10.1136/bmj.m441.

Reviewer #3 (Remarks to the Author):

Understanding the heterogeneity in disease phenotypes and endotypes is important for optimal treatment for individual patients. Here, the authors studied the molecular signature associated with clinical phenotypes of atopic dermatitis (AD) by integrating RNA-seq data of both skin and PBMC along with clinical data from 115 AD patients and matched 14 healthy controls. By combining weighted gene co-expression network analysis (WGCNA) and elastic net regression, the authors identified phenotypes-relevant transcriptome modules in skin and PBMC. Based on longitudinal features of blood tests and PBMC transcriptome modules, the authors identified three patient clusters, which were associated with longitudinal features in clinical severity as well as in the medication history.

One of the main conclusion in this study was the important contribution of the identified transcriptome modules to the clinical phenotypes. However, the regression model did not show clear improvement of prediction when adding transcriptome modules as model variables. In many places, method details were not clear. Below are specific comments.

Response:

We sincerely thank the reviewer for taking time to consider our manuscript and kindly acknowledging the value of our study on phenotype-endotype association in AD. We appreciate many insightful comments to further improve the manuscript.

Major points:

1. Fig. 6: what is the difference between linear model and elastic net model? Although the authors claimed better prediction when adding skin and PBMC modules, the improvement is very subtle (0.41 vs. 0.43 for R^2). It is not surprising to see such improvement. This result was not sufficient to draw the conclusion that transcriptome modules was closely related with phenotypes. The results still indicate that blood test was the dominant contributor.

Response:

Thank you for the insightful comments. As the reviewer pointed out, we examined the performance of both linear model and elastic net model on the same dataset to confirm unbiased choice of the analytical methods. Both are the regression methods where a dependent variable and independent variables are assumed to have a linear relationship. Their difference is that elastic net is penalized for weight choice while linear model is not penalized. Empirically, linear model tends to result in overfitting as model complexity increases, and this unfavorable effect can be alleviated by penalization. Our results of regression models for EASI prediction turned out to be following this rule (adj R^2 (training) = 0.65, adj R^2 (test) = 0.02 for linear model vs adj R^2 (training) = 0.64, adj R^2 (test) = 0.43 for elastic net model, when all the variables are used). We demonstrated superiority of the elastic net over linear model for application on our dataset in **Fig. 6a**.

The principal reason why we used both skin and PBMC modules in the regression model is that we intended to explore AD pathology on a molecular basis by utilizing biological subsystems yielded by gene expression. Moreover, pathological association of more than one tissue would provide insights into systemic regulation in pathology which had never been achieved by the previous studies with single omics analysis. Even though the contribution of

skin and PBMC modules were not relatively large in terms of model performance, this limitation can be mitigated by the conceptual advantage of our study.

Modification:

For the better clarification, we modified the description as follows:

“We next investigated how the AD phenotypes can be represented by transcriptome modules of both skin and PBMC, as well as by laboratory tests obtained at the same visits. Given the relatively large number of variables to the sample size, we built regression models using elastic net, an algorithm for regularized regression and variable selection that is applicable to high dimensional data with multicollinearity [37].

To confirm that regularized regression is superior to linear model in building regression models on our complex dataset consisting of both skin and PBMC transcriptome, we compared the performance of the linear model and elastic net (**Fig. 6a**). We found that adj R^2 for the test dataset is higher in elastic net model compared to linear model (adj R^2 (training) = 0.65, adj R^2 (test) = 0.02 for linear model vs adj R^2 (training) = 0.64, adj R^2 (test) = 0.43 for elastic net model, when all the variables are used), verifying the advantage of elastic net in our data. Although addition of transcriptional modules did not improve the overall model performance drastically, several gene modules that were selected as the predictor variables for the AD phenotypes provided insights into transcriptional regulation involving pathology.

Using elastic net, we also built models to predict EASI (erythema) and EASI (papulation) which made a major distinction in skin manifestations as described above. The model performance (adj R^2) and the set of significant features ($p < 0.05$) in each model as well as its biological characteristics are summarized in **Table 1.**” (**Results**, line 286)

2. It is interesting that the authors identified AD-related modules by applying WGCNA to the entire datasets including both AD patients and health control? Can the authors further discriminate the modules related to AD from modules related to health control?

Response:

Thank you for the insightful comment. As the reviewer mentioned, we identified AD-related modules by applying WGCNA to the entire datasets including AD patients and healthy controls (HC). According to the reviewer’s suggestion on discriminating the modules between AD patients and healthy controls, we applied WGCNA on subsets of the data of either of AD patients or HCs. We found that most of the gene modules identified in our original dataset were also defined in the AD-specific data, but only about half of the modules were defined in the HC-specific data, both in skin and PBMC.

Among the 21 modules identified in entire skin data, 9 modules were commonly defined in both AD and HC data. Therefore, they were considered to be general modules in skin tissue. In contrast, 10 modules were defined only in the AD data, which were accordingly considered to be AD specific. The general skin modules include modules specifically expressed by keratinocyte, inner root sheath/sebaceous gland (IRS/seba) and sweat gland, suggestive of metabolic homeostasis in skin tissue. On the other hand, AD specific modules include modules specifically expressed by T cell and myeloids along with keratinocyte, especially the modules that are characterized by immune-related GO terms; “Cytokine signaling”, “Innate immune system”, “Interferon signaling” and “Immune system”.

In PBMC data, 9 modules were found to be general in PBMC, while 5 modules were found to be AD specific. The general PBMC modules consisted of modules expressed by multiple cell types of both lymphocyte and myeloids with GO terms such as “Platelet activation” and “B cell receptor signaling”. On the other hand, AD specific modules include modules that are expressed by neutrophil, monocyte and Treg with GO terms such as “Neutrophil degranulation” and “FGFRs signaling”.

Those findings highlight dependency of module identification on data composition, thereby superiority of using complex data including AD patients and HCs in our analysis for identification of modules that are then used to illustrate population heterogeneity.

Modification:

To reflect the reviewer’s comments, we modified the description as follows:

“**Fig. 5b** and **5d** show the size of the first principal component (PC1) value per cell type obtained by applying principal component analysis (PCA) on gene expression data of cell types for each gene module (i.e. matrix with m columns of gene and n rows of cell types, where m is the number of genes assigned to a given module). See **Fig. S10** and **Supplementary note** for further characterization of the gene modules.” (**Results**, line 256)

“To confirm specificity of the identified modules in AD population, we separately applied WGCNA to the data subsets consisting of the AD patients and HCs (N; AD = 260, HCs = 55 for skin, and AD = 194, HCs = 41 for PBMC), and compared the newly defined modules with the original modules. These modules defined in two different datasets were then named sAD and sHC for skin modules, and pAD and pHc for PBMC modules, respectively (**Fig. S10 a, d**).

Similarity of the newly defined modules to the original modules was evaluated using Jaccard index with the following formula; Jaccard index (X, Y) = $|X \cap Y| / |X \cup Y|$, where X and Y represent the genes assigned to a given module defined in original dataset (sModu and pModu), and a given module defined in subset of data (sAD/sHC and pAD/pHC), respectively (**Fig. S10 b, e**). Newly defined modules with Jaccard index greater than 0.20 were deemed as equivalent modules to the original modules.

In the skin data, nine modules were defined in both AD and HC data, and therefore they were considered to be general modules in skin tissue, while 10 modules were defined only in AD data, which were accordingly considered to be AD specific. The general skin modules include modules specifically expressed by keratinocyte, inner root sheath/sebaceous gland (IRS/seba) and sweat gland, suggestive of metabolic homeostasis in skin tissue. On the other hand, the AD-specific modules include those specifically expressed by Tcell and myeloids along with keratinocyte, especially those characterized by the immune-related GO terms; “Cytokine signaling”, “Innate immune system”, “Interferon signaling” and “Immune system”.

In the PBMC data, nine modules were general in both AD and HC, while five modules were found to be AD specific. The general PBMC modules consisted of modules expressed by multiple cell types of both lymphocyte and myeloids with GO terms such as “Platelet activation” and “B cell receptor signaling”. On the other hand, AD specific modules include modules that are expressed by neutrophil, monocyte and Treg with the GO terms such as “Neutrophil degranulation” and “FGFRs signaling”.

Consistently, expression intensity was significantly higher in AD compared to HCs in the AD specific modules (**Fig. S10 c, f**). In contrast, expression intensity was greater in the HCs than the AD patients in sModu09 (GO: Extracellular matrix organization, top genes: *PI16*, *FBLN1*, *ADH1B*, *MFAP4*, *CFD*), sModu17 (GO: Formation of the cornified envelope, top genes: *FLG2*, *LOR*, *LCE5A*, *FLG*, *IL37*) in skin, suggesting the importance of these modules for normal function of epidermal barrier and connective tissue in healthy skin.” (**Supplementary note**)

a

b Similarity to original gene modules

c

d

PBMC modules identified in original dataset (AD+HC, N = 235)

Top 5 PC1 contributing genes

NELL2, LRRN3, OBSCN, CCR7, GPRASP1
FGFBP2, GZMH, GNLY, FCRL6, ADGRG1
PPBP, TUBB1, ITGB3, SDPR, SPARC
MKI67, RRM2, TOP2A, ASPM, MYBL2
SIGLEC1, IFI44L, SERPING1, IFIT3, OTOF
FCRL1, MS4A1, PAX5, CD22, LINC00926
VCAN, CYP1B1, FAM198B, CREB5, CD163
CFD, TYROBP, GPBAR1, KLF12, TSPO
S100A9, S100A12, PLBD1, CD14, PADI4
ALAS2, SLC4A1, CA1, SELENBP1, HBD
CCR4, CNTNAP1, DUSP4, LMNA, PI16
ATP5EP2, CXCL14, LOC100506675, ABCA8, LOC100287792
JCHAIN, IGLL5, MZB1, FAM46C, TNFRSF17
XIST, DDX3Y, TXLNGY, KDM5D, USP9Y
GATA2, CLC, HDC, MS4A3, MS4A2

PBMC modules identified in AD data (N = 194)

MKI67, RRM2, TOP2A, JCHAIN, ASPM
NELL2, LRRN3, CCR7, OBSCN, GPRASP1
GZMH, FGFBP2, FCRL6, ADGRG1, GNLY
VCAN, FAM198B, CYP1B1, CREB5, CD36
SIGLEC1, IFI44L, SERPING1, OTOF, IFIT3
FCRL1, MS4A1, PAX5, CD22, LINC00926
KLF12, GPBAR1, TSPO, FAM169A, MGAT4A
S100A8, S100A9, MCEMP1, S100A12, PLBD1
CCR4, CNTNAP1, DUSP4, PI16, LMNA
ATP5EP2, CXCL14, LOC100506675, ABCA8, LOC100287792
XIST, DDX3Y, KDM5D, TXLNGY, USP9Y

PBMC modules identified in HC data (N = 41)

S100A12, PLBD1, AQP9, S100A9, CYP1B1
GZMH, GNLY, ADGRG1, B3GAT1, FGFBP2
ALAS2, SLC4A1, TRIM58, SELENBP1, HBD
LRRN3, TRABD2A, OBSCN, NELL2, LEF1
PPBP, SPARC, TUBB1, SDPR, ITGB3
MTRNR2L1, KIAA0125, POU2AF1, CD79A, FCRL1
ANP32A.IT1, CELF2.AS1, ZBTB37, HMBOX1, SYNE3
TXLNGY, USP9Y, XIST, DDX3Y, KDM5D
MKI67, RRM2, MYBL2, TOP2A, ASPM
IFI44L, SIGLEC1, IFIT3, IFIT1, IFI44
JCHAIN, IGLL5, MZB1, FAM46C, TNFRSF17
LINC00641, CLK1, GPR18, ADRB2, ID3
LOC200772, CDKN1C, FCGR3B, FCGR3A, LINC01272

gdT: gamma delta T cell
 MAIT: mucosal associated invariant T cell

Fig. S10 Application of WGCNA on the AD/HC subset of data highlight differential patterns of AD-specific modules and generally observed modules in the tissue.

a,d. Cell type specific expression (left) and the list of top 5 PC1 contributing genes (right) of skin (**a**) and PBMC (**d**) modules identified in original dataset (upper), subset of data from AD patients (middle), and subset of data from HCs (lower). Cell type specificity was assessed by referring external dataset of skin single cell-RNA-seq (**a**) and sorted blood cell type RNA-seq data (**d**). **b,e.** Plots showing similarity between the modules defined in original dataset and modules defined in AD or HC dataset as evaluated by Jaccard index in skin (**b**) and PBMC (**e**). **c,f.** Histogram showing distribution of expression intensity of modules in AD patients (blue) and HCs (red) in skin (**c**) and PBMC (**f**). The classification of *p*-values of student's *t*-test or Welch's *t*-test (according to the homoscedasticity examined by *F*-test) between AD patients and HCs were described inside the brackets. **** $p < 0.0001$, *** $p < 0.001$, ** $p < 0.01$, * $p < 0.05$, NS: non significant. HC: healthy control.

3. Cell types was estimated using scRNA-seq data of AD patients and bulk PBMC RNA-seq. There are plenty of PBMC single cell datasets, Why not use it in Fig 5? In addition, why do not use the same reference cell types of PBMC in fig 4a and fig 5d?

Response:

Thank you for the insightful comment. As the reviewer pointed out, we estimated cell types responsible for expression of a given gene or module by referring to the external datasets; single-cell RNA-seq data of skin tissue for skin data, and bulk RNA-seq data of 18 cell types sorted from human peripheral blood (Human Protein Atlas: HPA, blood cell gene data) for PBMC data.

The reason why we used the HPA data for PBMC gene expression reference is that the experimental technique for cell type separation in blood sample is well established, as opposed to the skin tissue where cell type separation is still challenging. And therefore, the quality of bulk RNA-seq data of cell types sorted from blood is generally quite high with equally sufficient read number across cell types. In contrast, strategies for cell type annotation in PBMC sc-RNA-seq still remain open to discussion, especially for detailed annotation of T cell subsets and granulocytes. Indeed, inclusion rates of granulocytes into the PBMC fraction is altered dependent on subtle differences in experimental conditions of the cell separation process. In addition, unbalanced population of each PBMC cell type may compromise data quality of rare cell types such as Treg and pDC.

Considering those factors, we expected the HPA data as the best reference data for annotation of our PBMC data which contained unknown ratio of granulocytes. Actually, by referring HPA data, our dataset appeared to be containing substantial amount of neutrophil and basophil as expected.

Nonetheless, to answer to the reviewer's question clearly, we now conducted additional cell type estimation analysis using PBMC sc-RNA-seq data of 31 healthy Japanese subjects (NamKoong data, Namkoong et al., 2022). Following the annotation pipeline originally published by the data provider, we estimated the relative expression of the PBMC module genes among 15 cell types (CD14p_Mono, CD16_Mono, cDC, pDC, CD4_T, CD8_T, Treg, MAIT, Progenitor_T, NK, B, Plasmablast, Platelet, **Figure exclusively for the reviewer**). We found differential annotation results of the PBMC modules between the datasets. The most prominent difference was that modules that were annotated to basophil or neutrophil with the HPA data reference were annotated to platelet or pDC with the NamKoong data.

Although there may not exist the best answer to which data is most reliable as a reference for our data, some experimental evidence, e.g. *GATA2*, *CLC* and *HDC*, the top genes in pModu15, are important for basophil activity, support suitability of the HPA data for our situation.

[REDACTED]

(In addition, why do not use the same reference cell types of PBMC in fig 4a and fig 5d?)

As the reviewer pointed out in the latter part of the question #3, we did not use the same reference cell types of PBMC in **Fig. 4a** and **Fig. 5d**. Their major difference was that granulocytes were included in **Fig. 5d** but not in **Fig. 4a**. This is because of the differential assumption on how much contribution granulocytes have on the expression of targeted genes in two data, even though two data were originated from the same gene expression data.

In **Fig. 4a**, we focused the genes involved in cytokine – receptor coupling (i.e., 119 genes in PBMC), and expression of those genes were widely shared across multiple cell types. We annotated the cell types responsible for expression of each gene based on z-score > 2 in expression data which was normalized across cell types without weighting by cell frequency. With the presumption that the frequency of granulocyte is far less compared to other cell types, we considered that annotating these genes to cell types including granulocytes would lead to misinterpretation of cell type contribution to cytokine/receptor genes. We therefore considered that annotating cell types excluding granulocytes would be appropriate.

On the other hand, we investigated a wider range of genes that were assigned to gene modules in **Fig. 5d**. Some of these genes were highly specifically expressed by granulocytes (e.g., as mentioned above, *GATA2* and *HDC*, the top genes in pModu15, were almost exclusively expressed by basophil according to the HPA database). In this case, if the genes were annotated using cell type expression data excluding granulocytes, then the figure would become misleading. Therefore, we included granulocytes for cell type annotation for PBMB

module genes, although we still expect readers to have an image on how small the cell frequency of granulocytes is in PBMC upon interpreting this figure.

Modification:

To reflect the reviewer's comments, we modified the description as follows:

“For each of the cytokine and receptor genes, cell types responsible for the cytokine/receptor gene expression was estimated by referring to publicly available datasets (GSE147424; single-cell RNA-seq of skin tissue from AD patients and healthy controls [33], Human Protein Atlas blood cell gene data; RNA-seq of 18 cell types sorted from human peripheral blood [34], for skin and PBMC RNA-seq data, respectively). Reference datasets were standardized among cell types and genes that were expressed at a level of z-score > 2 were deemed as cell-type specific genes. Note that expression of cytokine/receptor genes were widely shared across multiple cell types in PBMC. Since contribution of granulocytes may be negligible because of their small fraction in PBMC compared to other cell types, we excluded neutrophil, eosinophil and basophil from the cell type annotation in this analysis. Ligand-receptor connection were visualized using the R package circlize [68].” (**Methods**, line 601)

“Module characterization was performed based on two terms, cell type specificity and GO. Cell type specificity in its expression was determined by referring to the same external dataset used in the previous section, i.e. either sc-RNA-seq (skin) or sorted cell RNA-seq (PBMC). Because number of genes in the PBMC modules was specifically expressed by granulocytes, we included neutrophil, eosinophil, and basophil for the cell type annotation in this analysis. Note that the cell type frequency was not taken into account for the size of the contribution to expression of each gene. GO analysis were performed with the R package clusterProfiler.” (**Methods**, line 623)

4. The time series features used in this study were single values, which cannot well characterize the dynamic changes over time. Why not use metrics (such as Pearson correlation) that can directly capture the relationship between patients based on the time series data? I am not saying that Pearson correlation is the best metric, but the authors should explore other metrics.

Response:

Thank you for the valuable comment. As the reviewer pointed out, we conducted dimension reduction of the time series data by extracting time series features such as mean and MAC in individual patients. Please note that time series data with maximum 12 time points per patient and month would capture clinical features involving the magnitude of the change in individual patients. While ours may not be fully enough sample sizes to conduct the typical pattern analysis such as Pearson's correlation to obtain quantitative similarity (or dissimilarity) across longitudinal trajectories, we consider the current data and analytic results may have essential clinical and biological meanings.

According to the reviewer's suggestion, we conducted linear mixed model (LMM) analysis to examine to what extent the time series data of blood derived parameters can explain the disease severity of individual patients in the longitudinal settings. We then examined whether personalized effects highlighted in LMM can be used for patient clustering.

Under the assumption that individual patients harbor their own random effects both in intercept and coefficient, LMM identified four blood derived parameters, TARC, Eosinophil, LDH and PLT as having significant fixed effects in linear relationship with disease severity. Several other factors, such as CRE and pModu11, showed substantial variance in the random effects of coefficient across patients (**Fig. S13a**). This suggested that these analytes could potentially be used as patient stratification factors.

We thus conducted patient clustering using the parameters that have large SD values of the random effect coefficients as well as parameters that have significant fixed effect (**Fig. S13b**). The patients were largely stratified into two subsets; patients who have large random effect intercept and small random effect coefficient, and those who have small random effect intercept and large random effect coefficient. The size of the random effect coefficient was inversely related to the size of random effect intercept in most of the parameters, especially in those with significant fixed effects (TARC, Eosinophil, LDH and PLT), RBC, and pModu11.

This patient clustering was similar to the original ones in **Fig. 8c**; parts of the patients were classified into the same group of the two types of clustering. We did not find clear relevance to clinical features in this patient clustering as opposed to clustering based on the time series features as shown in **Fig. 8**. We note that there existed some sex-specific differences in LMM analysis that might be linked with sex-specific clinical characteristics (e.g. the baseline level difference of RBC between males and females). This point should be investigated in the future study with increased sample size.

Modification:

To reflect the reviewer's comments, we modified the description as follows:

“These observations suggest that the predominant features vary by patients, which could limit the performance of linear models assuming same feature weights across samples. Application of linear mixed model (LMM) on each analyte in the time series data also highlighted the varying random effects by patients (**Fig. S13, Supplementary note**).” (**Results**, line 347)

“To examine the relationship between each of the blood analytes and the disease severity in individual patients in the longitudinal settings, we applied linear mixed model (LMM) on time series data using lmer() function from the lme4 R package [Bates et al., *J Stat Softw*, 2015]. Respective blood derived parameter was tested for linear relationship with disease severity, with patient IDs used as random effects. While fixed effect coefficient with $p > 0.05$ was considered to be significant, random effect coefficient with $SD > 0.13$ was considered to be at large variance across patients, suggested to be potential stratifying factors that would highlight the differential longitudinal features among patients.

Accordingly, four blood derived parameters, TARC, Eosinophil, LDH and PLT were found to have significant fixed effects in linear relationship with disease severity (coefficient $p < 0.05$), under the assumption that individual patients harbor their own random effects both in intercept and coefficient. Above all, TARC showed the smallest p -value and the largest coefficient value as a fixed effect, with substantial variance in random effects of both intercept and coefficient. This suggests that TARC bears varying size in both the baseline and effects on linear relation to disease severity by patients, thereby being helpful for fundamental characterization of patients in the context of inter-individual difference as well as intra-individual variation in the longitudinal setting.

Although other analytes did not show significant fixed effects in the relationship to disease severity, some of them such as CRE (creatinine) and pModu11 showed substantial variance in the random effects of coefficient across patients. In those analytes, coefficients of relationship between disease severity were found to vary largely by patients (**Fig. S13a**), which compromise the credibility in explanation capability of the values when data from all the patients were pooled. (Note that pModu14 which represents X- or Y-chromosome linked genes is an exception, since high level of SD was obviously produced because of outliers.) This feature of parameter can be advantageously used for patient stratification since it can highlight the differential longitudinal features among patients, and therefore, we conducted patient clustering using these parameters that have SD values of random effect coefficients above 1.3 (**Fig. S13b**).

Accordingly, we found that patients were basically stratified into two subsets; patients who have large random effect intercept and small random effect coefficient, and patients who have small random effect intercept and large random effect coefficient. It also became apparent that the size of random effect coefficient was inversely related to the size of random effect intercept in most of the parameters, especially in the parameters that have significant fixed effects (TARC, Eosinophil, LDH and PLT) as well as RBC and pModu11. This patient clustering is in part similar to the one demonstrated in **Fig. 8c** but no clinical relevance was found so far.” (**Supplementary notes**)

Fig. S13 Application of linear mixed model on time series data highlighted the varying random effects by patients.

a. Respective blood derived parameter was tested for linear relationship with disease severity, with patient IDs used as random effects. The data was plot with different colors according to the patients. The lines represent predicted values in each patient using the LMM with random effects. **b.** Random effects of both intercept and coefficient in selected parameters (fixed effect coefficient p-value < 0.05 and/or random effect coefficient SD > 0.13) in individual patients. coef: coefficient, int: intersect.

Minor points:

1. What is the meaning of endotypes in this study? In line 407, the authors mentioned that it was defined by the transcriptome modules.

Response:

Thank you for the insightful comment. In general, endotype refers to the subtypes with distinct biological mechanisms that involves identification of molecular signatures by analyzing the patient samples. The signatures could be defined based on various forms of output in any omics analysis such as transcriptomics and metabolomics. In our study, we defined them based on tissue RNA-seq analysis among other omics analysis as represented by skin modules and PBMC modules.

Modification:

To reflect the reviewer’s comments, we modified the description as follows:

“Our combinatorial approach of WGCNA and elastic net regression enabled us to efficiently and jointly analyze high dimensional datasets of skin and PBMC transcriptomes. Our finding on skin manifestation-dependent molecular profiles suggests that endotypes in AD (i.e. biological subtypes that were defined based on tissue transcriptome analysis in our study) are closely associated with the phenotypes of AD that were defined by visual evaluation of skin. More fundamentally, this observation supports the assumption that the AD population comprises a variety of pathophysiological subtypes.” (**Discussion**, line 426)

2. Fig 6b: Were the predictors such as Lymphocyte and Eosinophil included in the elastic net model? What are the input values of these variables? What are the meanings of the links in Fig. 6b?

Response:

Thank you for the practical comments. As the reviewer pointed out, contents of the blood test variables used in the elastic model were not clearly indicated in the manuscript. Therefore, we now added the list of blood test variables in **Table S6**. These variables were used in the elastic models after standardization across study subjects including both AD patients and healthy controls, as indicated in the yellow box of “Blood tests” in **Fig. 6a**. In **Fig. 6b**, all the selected variables in elastic net were indicated with circle with different colors according to their sample categories (i.e. skin or PBMC or blood).

Modification:

To reflect the reviewer’s comments, we modified the description as follows:

“We next investigated how the AD phenotypes can be represented by transcriptome modules from both skin and PBMC, as well as by laboratory tests (**Table S6**) obtained at the same visits.” (**Results**, line 286)

Table S6 A list of blood test variables used in regression models. Individual variables were standardized across study subjects. WBC: white blood cell, RBC: red blood cell, LDH: lactate dehydrogenase, PLT: platelet, CRP: C-reactive protein, HGB: hemoglobin, TB: total bilirubin, BUN: blood urea nitrogen, CRE: creatinine, AST: aspartate aminotransferase, ALT: alanine transaminase, GGTP: gamma-glutamyl transferase.

Category	Variable	Unit	Mean ± SD
Hemogram	WBC	10 ³ cell/μL	6.73 ± 1.73
Hemogram	RBC	10 ⁶ cell/μL	4.86 ± 0.51
Hemogram	Lymphocytes	%	24.41 ± 7.84
Hemogram	Neutrophil	%	62.71 ± 8.33
Hemogram	Eosinophil	%	6.01 ± 4.71
Hemogram	Basophil	%	0.68 ± 0.37
Hemogram	Monocyte	%	6.09 ± 1.74
Biochemistry	LDH	U/L	235.50 ± 75.14
Biochemistry	PLT	10 ³ cell/μL	271.96 ± 57.31
Biochemistry	CRP	mg/dL	0.16 ± 0.36
Biochemistry	HGB	g/dL	14.52 ± 1.59
Biochemistry	TB	mg/dL	0.71 ± 0.34
Biochemistry	BUN	mg/dL	13.69 ± 2.83
Biochemistry	CRE	mg/dL	0.80 ± 0.15
Biochemistry	AST	U/L	23.06 ± 8.72
Biochemistry	ALT	U/L	26.69 ± 21.18
Biochemistry	GGTP	U/L	29.26 ± 20.08
Immunoassay	Total IgE (log2)	-	10.20 ± 3.17
	(Total IgE (raw))	IU/mL	4903 ± 6710
Immunoassay	TARC (log2)	-	10.02 ± 1.64
	(TARC (raw))	pg/mL	2348 ± 6009

3-1. Fig. 5b: PC1 value was computed for each module per patient. How were the values related to cell types as shown in Fig. 5b?

Response:

Thank you for the insightful comment. As the reviewer mentioned, two types of PC1 values were computed; (1) the PC1 value computed for each module *per patient* to separately characterize each patients, and (2) the PC1 value computed for each module *per cell type* to characterize each module. Please note that these two types of PC1 values were computed using two different matrices. The former was conducted on the matrix showing gene expression of individual modules in individual patients, and the latter was conducted on the matrix showing gene expression of individual modules in individual cell type. **Fig. 5b and d** are plots showing the size of the PC1 value *per cell type*, which represent relative cell type specificity of module genes across cell types. To clarify distinction between the size of PC1 value *per patient* and the size of PC1 value *per cell type*, we added the plots showing the size of PC1 value *per patient* as well in **Fig. S9a**.

Modification:

To reflect the reviewer's comments, we modified the description as follows:

"As expected, genes in each module exhibited substantial cell type specificity in their expression as confirmed by referring to the publicly available dataset of either single-cell

RNA-seq (scRNA-seq) (for skin) or sorted cell RNA-seq (for PBMC). **Fig. 5b** and **5d** show the size of the first principal component (PC1) value per cell type obtained by applying principal component analysis (PCA) on gene expression data of cell types for each gene module (i.e. matrix with m columns of gene and n rows of cell types, where m is the number of genes assigned to a given module).” (**Results**, line 254)

“To obtain personalized profiles based on the transcriptional modules, scores for each module and each patient were defined. Since identified modules consist of co-expressing genes, expression patterns in each module became simple enough to be handled linearly, as verified by substantially high value of explanatory capability of PC1 (40-60%) when PCA was applied on gene expression data of patients for each gene module. Therefore, we used the PC1 values followed by standardization across patients as the index of intensity of gene expression of transcriptome modules in each patient (**Fig. S9**).” (**Results**, line 276)

Fig. S9 Statistics of transcriptome modules associated with AD.

a. A plot shows the size of PC1 value per patient. PC1 value were obtained by applying PCA on gene expression data of patients for each gene module.

3-2. It was interesting that sModule2 with high PC1 values in keratinocytes was enriched in neuron system. What did this mean in biology?

Response:

Thank you for the insightful comment. With regard to the enrichment of GO term “neuron system” in sModu02, we hypothesized that sModu02 is the signature representing somatosensory receptors in skin tissue such as Pacinian corpuscle and Ruffini ending. The supporting evidence for this hypothesis is that sModu02 includes *KCNH4*, *CACNA1A* and *ASIC2*, genes coding potassium voltage-gated channel subunits, calcium voltage-gated channel subunit, and acid sensing ion channel subunit, respectively, all of which is known to be expressed in sensory neuron. Among those genes, *ASIC2* had been demonstrated to be specifically expressed in Pacinian corpuscle in human skin (Calavia et al., 2010; Geffeney and Goodman, 2012), which supports our inference of sModu02 as a representation of somatosensory receptor. In the future, more comprehensive analysis with spatial RNA-seq may prove our hypothesis on a microstructure specific gene signature in skin.

Modification:

To reflect the reviewer’s comments, we modified the description as follows:

“Additionally, three modules were found to be representing skin appendages; sebaceous gland (sModu01, GO: fatty acid metabolism) and sweat gland (sModu03 and sModu19, GO: ion channel transport and developmental biology, respectively). The intensity of these modules was not relevant to dermatitis, and was strongly biased by sampling regions. Therefore, we considered these two modules as noises, and excluded from the following analysis. Another potential representation of skin appendage, although it is not evident as much as above mentioned three modules, is a neuroreceptor signature by sModu02 which include *KCNH4*, *CACNA1A* and *ASIC2*, genes coding ion channel subunits with suggested association with sensory neuron in human skin [36].” (**Results**, line 267)

4. It is not clear how did the PCA was performed in Fig. S4? What is the input of the PCA analysis?

Response:

Thank you for pointing out this important point. As mentioned in the answer to the previous comment, PCA in **Fig. S9** was performed on the expression data matrix (in vst value) for each gene module, i.e. m columns of gene and n rows of patient, where m is the number of genes assigned to the given module. Therefore, we modified the manuscript as shown in the previous comment.

5. Fig. 7: It is confusing to show the axis on the most left. I suggest the authors to show the x/y-axis labels along with the dot plots and only leave the figure legend on the most left panel. The authors mentioned “R2: 0.10 vs. 0.26” in line 320, but these two numbers are not the same as in Fig. 7a. I am very surprised that the calculated R2 on the test samples in basic information + blood test was negative when examining the scatter plot in Fig. 7a. Why do not consider skin modules?

Response:

Thank you for the practical suggestion. We modified the axis labels, figure legend and the description of R^2 in the manuscript line 320.

Regarding the reviewer’s comment on a negative value of R^2 on the test set, please note that R^2 can generally take negative values depending on situations.

R^2 itself is defined as:

$$R^2 = 1 - \text{SS.residual}/\text{SS.total},$$
$$= 1 - \frac{\sum(y_i - y_{i,\text{predicted}})^2}{\sum(y_i - \text{mean}(y))^2}$$

(SS: the sum of squares)

Given this definition, in the case where SS.residual exceeds SS.total, which means the sum of the errors of predicted values is greater than the variation of the actual values, negative R^2 will be produced. A model with positive R^2 in the training set and negative R^2 in the testing set may represent overfitting, as seen in our regression model with basic information + blood test (train $R^2 = 0.61$, test $R^2 = -0.24$). However, this potential overfitting issue was mitigated by adding PBMC modules as independent variables, which resulted in train $R^2 = 0.64$ and test $R^2 = 0.15$.

As the reviewer pointed out, we did not include skin modules in the regression model in the longitudinal dataset. It is because that the main purpose of this analysis is to find out how accurately can the time series blood data predict the individual disease course in the longitudinal setting. Time series acquisition of skin data is hardly feasible given the ethical issue arising from sampling invasiveness, and therefore, the major challenge in longitudinal analysis is to efficiently extract biology in individual patients using data with minimally invasive sampling. In addition, not all the patients in longitudinal cohort have skin data, which will further limit the sample size of this analysis. For these reasons, we excluded skin data from regression models.

Modification:

We modified the description of R^2 in the manuscript.

“Prediction performance for EASI (total) was higher in a model using all of basic information (age, age², sex), laboratory tests and PBMC transcriptome compared to a model using only basic information and laboratory tests (R^2 : 0.15 vs -0.24).” (Results, line 339)

6. Fig. 8: Why did the authors only consider the “Mean” and “MAC” when assessing the association of patient clusters with the variables?

Response:

Thank you for the insightful question. As the reviewer pointed out, we showed the plot of Mean and Mac as representative time series features in Fig. 8. As we mentioned in the manuscript, hierarchical clustering of those 7 features of clinical severity in 30 AD patients showed two major clusters; one includes mean, maximum, minimum and RMS, the other includes MAC, CID and approximate entropy (Fig. S14). The parameters in the same cluster were found to behave similarly across patients. Among the parameters from two clusters, we picked “mean” and “Mac” as representative values, because those parameters are widely used in various situation and intuitively interpretable for the reader.

Modification:

For the better clarification, we modified the manuscript as follows:

“Hierarchical clustering of those 7 features of clinical severity in 30 AD patients showed two major clusters; one includes mean, maximum, minimum and RMS, the other includes MAC, CID and approximate entropy (Fig. S14). Therefore, we picked mean and MAC as representative values in two clusters, respectively, for demonstration of feature distribution among patients.” (Results, line 363)

7. What is the alpha in line 592?

Response:

Thank you for asking this important point. Alpha in this manuscript indicates the hyperparameter in the elastic net that determines the ratio of L1 penalty to the combination of L1 and L2 penalties given to the regularized regression. The condition of $\alpha = 1$ means exclusive L1 penalty equivalent to the lasso, while $\alpha = 0$ means exclusive L2 penalty equivalent to the ridge. In our study, we used $\alpha = 0.5$, which means the equal weight of L1 and L2 penalties for regularization, intending to exploit both the sparse representation effect in the lasso and the grouping effect in the ridge.

Modification:

For the better clarification, we modified the manuscript as follows:

“Elastic net, a regularization and variable selection method that combines the L1 and L2 penalties of the lasso and ridge methods [37], was applied on cross-sectional datasets consisting of both skin and PBMC RNA-seq data along with blood tests (AD patients: $n = 115$,

healthy controls: $n = 14$) to determine the strength of the relationship between disease phenotypes and omics features in AD using the R package glmnet [71]. For each phenotype defined in compositional analysis of clinical scores, total samples labeled with the degree of specific skin conditions in continuous values were split into a training set (70%) and a testing set (30%). Models were built on the training set with optimization of the regularization parameter λ , which determines how much shrinkage is used to train the model, through ten-fold cross validation. Another hyperparameter of α , which determines the ratio of L1 penalty to the combination of L1 and L2 penalties was set to 0.5, intending to exploit both the sparse representation effect in the lasso and the grouping effect in the ridge. Then the model with the optimal parameters was applied to the test set to get the R^2 value to evaluate how well the model fit to the observed data.” (Methods, line 634)

References

- Calavia, M.G., Montano, J.A., Garcia-Suarez, O., Feito, J., Guervos, M.A., Germana, A., Del Valle, M., Perez-Pinera, P., Cobo, J., and Vega, J.A. (2010). Differential localization of Acid-sensing ion channels 1 and 2 in human cutaneous pacinian corpuscles. *Cell Mol Neurobiol* 30, 841-848. 10.1007/s10571-010-9511-2.
- Geffeney, S.L., and Goodman, M.B. (2012). How we feel: ion channel partnerships that detect mechanical inputs and give rise to touch and pain perception. *Neuron* 74, 609-619. 10.1016/j.neuron.2012.04.023.
- Namkoong, H., Edahiro, R., Takano, T., Nishihara, H., Shirai, Y., Sonehara, K., Tanaka, H., Azekawa, S., Mikami, Y., Lee, H., et al. (2022). DOCK2 is involved in the host genetics and biology of severe COVID-19. *Nature* 609, 754-760. 10.1038/s41586-022-05163-5.

REVIEWERS' COMMENTS

Reviewer #1 (Remarks to the Author):

No further comments.

Reviewer #2 (Remarks to the Author):

Authors have answered adequately to all of my comments.

I do not have anything else to add.

I suggest to publish revised version of the manuscript.

Reviewer #3 (Remarks to the Author):

The authors have well addressed my concerns. I have no further comments.